# An atlas of small non-coding RNAs in human preimplantation development

Stewart J. Russell[1,13], Cheng Zhao [2,3,13], Savana Biondic[4,5], Karen Menezes[1], Michael Hagemann-Jensen [6], Clifford L. Librach[1,7,8,9,10,11] & Sophie Petropoulos [2,3,4,5,6,12] ✉

Understanding the molecular circuitries that govern early embryogenesis is important, yet our knowledge of these in human preimplantation development remains limited. Small non-coding RNAs (sncRNAs) can regulate gene expression and thus impact blastocyst formation, however, the expression of specific biotypes and their dynamics during preimplantation development remains unknown. Here we identify the abundance of and kinetics of piRNA, rRNA, snoRNA, tRNA, and miRNA from embryonic day (E)3-7 and isolate specific miRNAs and snoRNAs of particular importance in blastocyst formation and pluripotency. These sncRNAs correspond to specific genomic hotspots: an enrichment of the chromosome 19 miRNA cluster (C19MC) in the trophecto-derm (TE), and the chromosome 14 miRNA cluster (C14MC) and MEG8-related snoRNAs in the inner cell mass (ICM), which may serve as 'master regulators' of potency and lineage. Additionally, we observe a developmental transition with 21 isomiRs and in tRNA fragment (tRF) codon usage and identify two novel miRNAs. Our analysis provides a comprehensive measure of sncRNA biotypes and their corresponding dynamics throughout human preimplantation development, providing an extensive resource. Better understanding the sncRNA regulatory programmes in human embryogenesis will inform strategies to improve embryo development and outcomes of assisted reproductive technologies. We anticipate broad usage of our data as a resource for studies aimed at understanding embryogenesis, optimising stem cell-based models, assisted reproductive technology, and stem cell biology.

Proper blastocyst formation is largely governed by a coordinated remodelling in transcriptomic, proteomic, and epigenetic landscapes, initiated shortly after fertilisation in the zygote (1 cell) until the formation of distinct lineages observed in the blastocyst[1].

Additionally important are signal transduction, clearing of maternal mRNA, and spatial/physical cues[2–5]. Our understanding of these coordinated and complex processes in early embryogenesis is expanding, however, those pertaining specifically to human

[1]CReATe Fertility Centre, Toronto, ON, Canada. [2]Department of Clinical Science, Intervention and Technology, Karolinska Institutet, Stockholm, Sweden. [3]Division of Obstetrics and Gynecology, Karolinska Universitetssjukhuset, Stockholm, Sweden. [4]Faculty of Medicine, Molecular Biology Program, Université de Montréal, Montréal, QC, Canada. [5]Centre de Recherche du Centre Hospitalier de l'Université de Montréal, Axe Immunopathologie, Montréal, Canada. [6]Department of Cell and Molecular Biology, Karolinska Institutet, 171 77 Stockholm, Sweden. [7]Department of Laboratory Medicine and Pathobiology, University of Toronto, Toronto, ON, Canada. [8]Department of Obstetrics and Gynaecology, University of Toronto, Toronto, ON, Canada. [9]Department of Physiology, University of Toronto, Toronto, ON, Canada. [10]Sunnybrook Research Institute, Toronto, ON, Canada. [11]Institute of Medical Sciences, University of Toronto, Toronto, ON, Canada. [12]Faculty of Medicine, Département de Médecine, Université de Montréal, Montréal, QC, Canada. [13]These authors contributed equally: Stewart J. Russell, Cheng Zhao. ✉e-mail: sophie.petropoulos@umontreal.ca

preimplantation development and lineage specification remain limited. Small non-coding RNAs (sncRNAs), typically shorter than 200 nucleotides, are comprised of several main classes of sncRNAs, including microRNAs (miRNAs), transfer RNAs (tRNAs) and their fragments, piwi-interacting RNAs (piRNAs), small interfering RNAs (siRNAs), rRNA fragments (rRNAs), and small nucleolar RNAs (snoRNAs). Each of these sncRNAs are known to play important roles in reproduction and development through regulation of the cell cycle, stem cell maintenance, and differentiation (reviewed in refs. 6,7). From them, miRNAs are particularly well characterised due to their relative conservation in function between species and cellular contexts; however their precise sncRNA expression kinetics (temporal and lineage-related) and function during human embryogenesis remain to be identified.

miRNAs are generated by either a canonical or less prevalent, non-canonical pathway(s) and generally consists of cleavage of precursor molecules (pri-miRNAs) by a microprocessing complex which includes DROSHA and DGCR8[8]. The cleaved pre-miRNA is then further processed by DICER and loaded onto the Argonaute (AGO) family of proteins[9]. Increasing evidence from stem cell models suggests that miRNAs may be important players in embryonic lineage specification and cell fate. Recently, a unique miRNA profile (miR-15b, miR-322, and miR-467g) in human trophoblast stem cells (hTSC) was demonstrated to target a host of conserved genes in ESCs, resulting in a trophoblast phenotype upon overexpression[10]. Further, the importance of the primate-specific chromosome 19 miRNA cluster (C19MC) cluster for hTSC differentiation was demonstrated, where C19MC was actively expressed in naïve, but not primed ESCs, and knockout of the C19MC locus abolished TSC differentiation capacity[11], supporting a regulatory role of miRNAs in lineage formation. Moreover, in hESCs, knockout of DICER1 (critical for miRNA and siRNA biogenesis) leads to cell apoptosis and failed self-renewal, with essential pro-survival roles identified for miRNA clusters miR-302-367 and miR-371-373[12]. In contrast, Dicer knockout in mouse ESCs, maintained morphology and marker expression despite an accumulation of G1 phase cells, however, depletion of Dicer in mouse TSCs consistently resulted in growth arrest and differentiation towards trophoblast giant cells, indicating that miRNAs play a crucial role in the self-renewal of mouse TSCs[13]. Together, these studies have demonstrated the importance of miRNAs in stem cell maintenance and lineage differentiation in vitro, while highlighting important species-specific differences between human and mouse models. We now speculate that miRNAs may play an important role in initiating or maintaining cell type in the human embryo.

To address these gaps in knowledge, we utilised Smallseq[14] in parallel with Co-seq[14,15] on single-cells derived from human embryos spanning embryonic day (E) 3 to E7. We determined that piRNA, rRNA, snRNA, snoRNA, tRNA, and miRNA were present during human preimplantation development and that the developmentally-relevant sncRNA biotypes were miRNA and snoRNAs, which increased with time suggesting de novo genesis. Further, we identified miRNA and snoRNA signatures associated with pre-lineage, ICM and TE cells, which reflect changes seen in previously published comparisons between naive and primed ESCs, and identified that the C14MC and C19MC genomic region may serve as 'master' regulators of lineage specification and maintenance. Additionally, we observed a developmental transition with 21 isomiRs and in tRNA fragment (tRF) codon usage and two novel miRNAs. These findings suggest that sncRNAs, miRNAs (both canonical and novel) in particular, may regulate developmentally-associated genes and in turn contribute to lineage formation or maintenance during early human embryogenesis. To our knowledge this is the first extensive report of all sncRNAs throughout preimplantation development, providing insights into their dynamics and potential regulatory roles in human blastocyst formation.

## Results

### Quantification of small non-coding RNAs across human pre-implantation development

To comprehensively evaluate the sncRNAome during preimplantation development and lineage segregation, we profiled single cells from human preimplantation embryos spanning embryonic days (E) 3 to 7 (E3-E7) (Fig. 1a). Female patient age ranged from 26-45, with an average of 36.6 years. Following quality control, we analysed 972 cells from 69 embryos with an average of 65.2k unique molecular identifiers (UMIs) annotated to known sncRNAs from available databases (see methods) at E3, decreasing to 29.3k at E6/7 (Fig. 1b, c, Supplementary Data 1), which was largely attributed to a decrease in piRNAs and tRNAs (Fig. 1c). The main biotypes of sncRNAs and their proportion were identified including miRNAs (9.62%), snoRNAs (23.8%), rRNA (18.6%), piRNAs (17.2%) and tRNAs (29.3%) (Fig. 1c). We observed that while piRNAs and tRNAs decreased with development, suggesting the majority are parentally inherited, miRNAs, snoRNAs, and rRNA fragments gradually increased (Fig. 1c) - a trend that has been reported to begin in human zygotes[16]. The length distributions from each class show typical patterns, with primary peaks at 22 nt for miRNA, 74 nt for tRNA, and 30 nt for piRNA (Fig. 1d), further verifying the quality of data obtained.

We next aimed to better understand the change in expression patterns observed for the sncRNA biotypes by determining the expression of key components involved in the biogenesis of sncRNAs, as these have not been previously reported across preimplantation development. To do this, we leveraged our published embryonic data which mainly focused on protein-coding RNAs[17]. We observed an overall increase in the pri-miRNA processing machinery (DGCR8, DROSHA), with a relative decrease in piRNA-interacting proteins and maturation factors (PIWIL2, PIWIL3, HENMT1) with development (Fig. 1e). The general pattern in expression of these components fits with the decrease in piRNA observed with development and further supports their lack of de novo generation. Further, the gradual increase in miRNA micro-processing complex and DICER supports the increase in miRNA molecules observed with development in the embryo. The Argonaute proteins AGO1-4 form the base for the RNA-induced silencing complex (RISC) and appear to fluctuate by developmental time and lineage, but are expressed quite abundantly, suggesting a functional role of miRNAs associated with blastocyst formation.

We also examined the genomic location of reads that do not overlap with the main types of small RNAs, by comparing the mapping positions with annotations for protein-coding regions, lncRNA, and repeat regions from Repbase[18], as well as other annotations from GENCODE[19] (predominantly pseudogenes) (Supplementary Fig. 1). The majority of these reads are primarily from the sense strand of annotated protein-coding regions (10.8%) and lncRNAs (9.8%), with read length peaks at 18 and 21 nt. Reads overlapping with repeat regions are mainly from annotated LTR retroposons (1.5% on the sense strand and 0.8% on the antisense strand), with a read length peak at 18 nt. It remains unclear whether these reads have a biological or technical origin.

### Identification of cell-type specific miRNA signatures

Our knowledge surrounding the role of sncRNAs is limited during mammalian preimplantation development, particularly in the human. We reasoned that if we could identify biotype signatures which correspond to cell type and determine their developmental dynamics, we may begin to unravel some additional mechanism(s) underlying blastocyst formation. Given the lack of known sncRNA markers for lineages in the human embryo, we used our previously developed method, Co-seq, to first accurately classify our cell types[14]. Co-seq relies on a split-cell experiment, followed by library amplification of sncRNA using Small-seq and RNA using Smart-seq2. Leveraging our previous

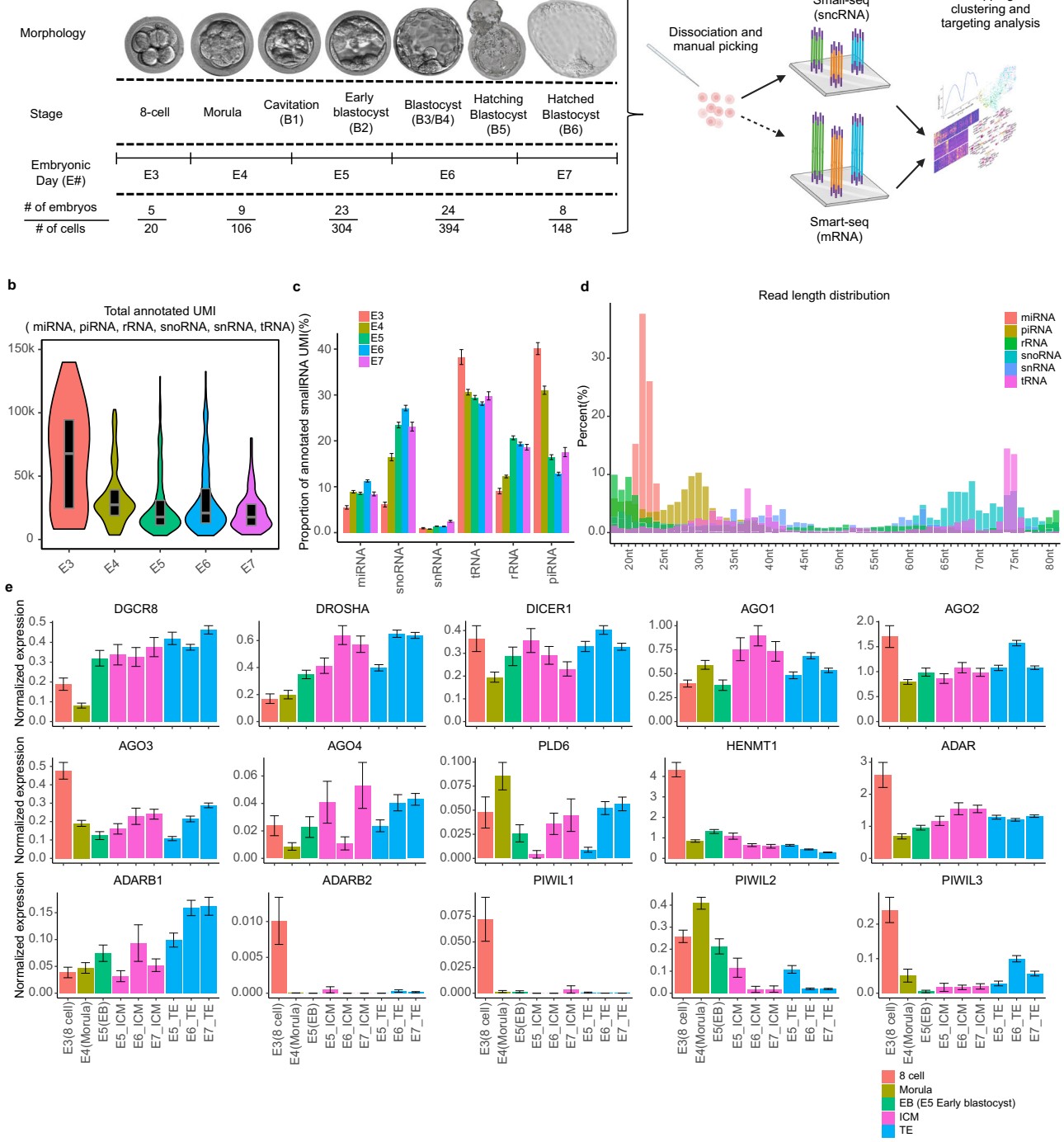

**Fig. 1 | Single-cell profiling of sncRNAs in human embryos. a** Overview of the experimental workflow. Donated embryos were cultured from E3-E7, dissociated into single cells, and manually picked for Small-seq or Co-seq. The dotted arrow indicates where a subset of cells were used. Number of cells and embryos remaining following QC filtering were attached below (created with BioRender). **b** Total UMIs and (**c**) proportion of UMIs annotated to microRNA (miRNA), small nucleolar RNA (snoRNA), small nuclear RNA (snRNA), transfer RNA (tRNA), ribosomal RNA (rRNA), and piwi-interacting RNA (piRNA) by day of embryonic development. The number of cells for each day is as follows: E3: 20, E4: 106, E5: 304, E6: 394, E7: 148. The boxplot rectangles represent the first and third quartiles, a horizontal line inside the box indicates the median value. **d** Percentage of total annotated reads by sncRNA class of various sequence lengths. **e** Gene expression of sncRNA processing genes across embryonic development and lineage (inner cell mass (ICM) and trophectoderm (TE)). Error bars represent standard error of the mean (SEM) values. The number of cells for each day is as follows: E3: 61, E4: 144, E5: 245, E6: 278, E7: 361. Created in BioRender. Kwok, Y. (2023) BioRender.com/v41c600.

single-cell RNA-sequencing (scRNA-seq) data, we integrated it with the current scRNA-seq from Co-seq to identify cell types observed in our Co-seq data (Fig. 2a, b)[17]. This allowed us to accurately classify the ICM and TE, in addition to the EPI, PE, polar and mural lineages (Fig. 2b, c). Using the Small-seq portion of each split cell, we were then able to overlay cell type with clusters observed and generate unique miRNAs

for each of these lineages. We observed the greatest enrichment of miR-376c-3p and miR-376a-3p in ICM cells and miR-525-5p and miR-518b in TE cells (Fig. 2d). Although relatively little is known about the role of the miR-376 cluster in embryo development, the functions of the chromosome 19 miRNA cluster (C19MC) which include miR-525 and miR-518b, have been well established as markers of TE differentiation[11].

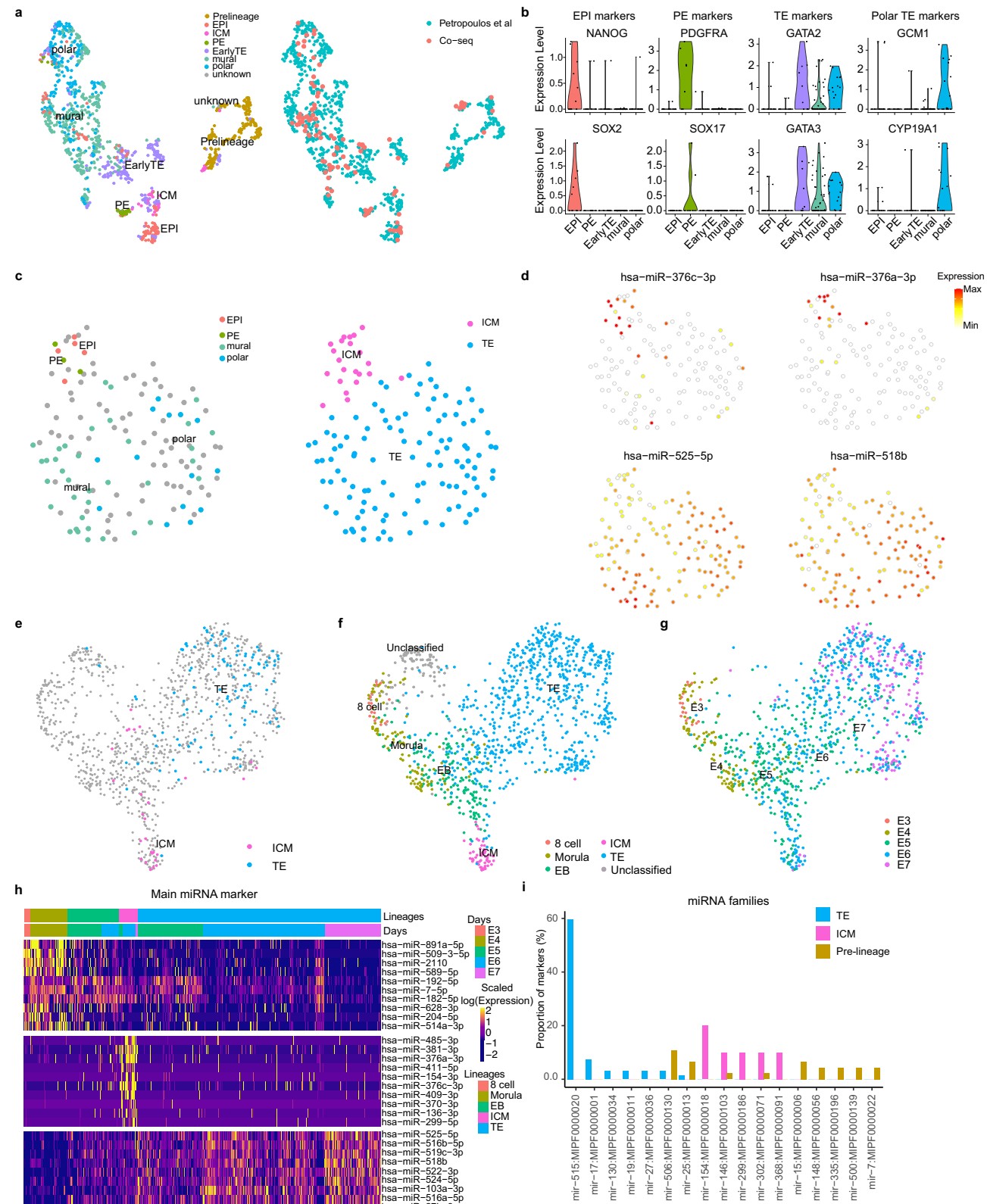

**Fig. 2 | Lineage identification and miRNA dynamics across embryo development.** **a** Integration of Co-Seq transcriptome data with[5] to establish cell identities. **b** Violin plots of marker gene expression in Co-Seq data. **c** UMAP of miRNA expression data from Co-Seq cells, coloured by cell identity and lineage. **d** Expression of marker miRNA miR-376c-3p (ICM), miR-376a-3p (ICM), miR−525−5p (TE), miR−518b (TE) in miRNA portion of Co-Seq data. **e** UMAP of miRNA expression in all cells passing QC, **f** coloured by lineage and (**g**) developmental time. **h** Heatmap of top significantly enriched miRNA expression in each lineage. **i** The proportion of miRNA markers in each miRNA family, split by lineage. EPI epiblast, ICM inner cell mass, PE primitive endoderm, TE trophectoderm, E3-E7 embryonic days 3–7, EB early blastocyst.

To verify the cell types present in our comprehensive dataset and identify sncRNA signatures for each cell type, we integrated the lineage-specific miRNA signatures obtained from the Co-seq experiment (Fig. 2e). Taking into consideration embryonic day, we identified miRNA expression dynamics from E3 to E7, which we found corresponded to pre-lineage (8-cell, morula, and early blastocyst (EB; just prior to ICM and TE specification)), followed by transition into ICM and TE (Fig. 2f, g). A cluster of unclassified cells was found to consist of cells from all developmental timepoints (Supplementary Fig. 2a–c), which were excluded from downstream analysis as they did not appear to be developmentally related. Developmental time contributed to the largest changes in miRNA expression; with lineage specific miRNA emerging at E5, just after cavitation, corresponding to the first lineage split of ICM and TE, and increasing to E7 (Fig. 2f, g). In addition, we investigated the relationship between embryo morphology (stage) and miRNA expression, and observed that miRNAs clustered more congruently with embryonic day (in which we also considered morphological parameters) (Supplementary Fig. 2d, e). We next identified those unique miRNA markers that corresponded to each lineage, with the top 10 displayed (Fig. 2h, Supplementary Data 2). We observed an enrichment of miR-506 family members in pre-lineage cells, including: hsa-mir-508-3p, hsa-mir-509-3p, hsa-mir-513a-5p, and hsa-mir-514a-3p (Fig. 2h, i, Supplementary Data 2). This cluster is a developmentally-related locus with high expression in spermatogenesis[20,21], however its function(s) in early embryogenesis are unknown. High levels of expression of the pluripotency-associated miR-182-5p and miR-183-5p were also detected in pre-lineage cells[22]. All of the top 10 enriched miRNAs in ICM cells, including miR-381-3p, miR-376c-3p and miR-299-3p, have been implicated in the maintenance of ground-state pluripotency in mouse ESCs and ICM (Fig. 2h)[22,23]. The majority of the TE-enriched miRNA we identified, including miR-518b, miR-519c-3p, and miR-27b-3p, are well-known trophoblast miRNA markers[24,25]. We now speculate that these miRNAs may be of particular importance given that they are now observed in the preimplantation TE and maintained during postimplantation. Perhaps, early dysregulation of these miRNAs during the preimplantation window may result in suboptimal placental formation and pathological conditions, providing possible targets for predicting embryo competence in clinic. We further investigated the marker miRNAs to determine their genomic origin. We observed that approximately ~60% of those identified in the TE belong to the miR-515 gene family, embedded in the C19MC (Fig. 2i). Similarly, the miR-154 family originating from a miRNA cluster on chromosome 14, is highly represented in the ICM lineage and contributes to ~20% of enriched miRNA.

Given that human ESCs are derived from the preimplantation embryo, we next asked whether the expression of embryonic miRNA markers observed in the pre-lineage, ICM, and TE were similar to naïve and primed ESCs, as well as the cancer cell line, HEK293T, as a control (Supplementary Fig. 3a). We observed no obvious enrichment of the pre-lineage markers in any of the three cell lines. However, a clear enrichment of almost all of the ICM and TE markers was observed in naïve ESCs, many of which have been shown to be required for naive pluripotency in ESCs[11]. There were three notable exceptions in the ICM markers, miR-302a-5p, miR-302b-5p, and miR-367-3p, all of which were higher in primed ESCs. We also did the reverse analysis and examined the expression of miRNAs determined to be specific to naïve and primed hESCs (Supplementary Fig. 3b), which confirmed an enrichment of naïve pluripotency miRNAs in the ICM. Naïve hESCs are known to be representative of the E6/E7 human preimplantation embryo and these data further corroborate the close molecular relationship between naïve hESCs and blastocyst derived cells.

The most striking finding from our enrichment analysis is the identification of two genomic hotspots associated with the lineage clusters: upregulation of a 123 kb region in TE cells located at Chr.19q13 (C19MC), and of a 180 kb region in ICM cells positioned at Chr.14q32

(C14MC; Fig. 3a, b). In the ICM, we observed 67% (14/21) of the marker miRNA originating from the sense strand of the C14MC cluster (Fig. 3a, b). We first detected C14MC expression in the early blastocyst, just prior to distinct ICM-TE lineage specification (Fig. 3c), and then remained highly enriched only in the ICM (Fig. 3c). The enrichment of the C14MC cluster exclusively in the ICM was surprising as it is reported in post-implantation trophoblast cells[26], similar to the C12MC cluster which is the mouse counterpart (reviewed in ref. 27). Mir-381-3p, originating from the C14MC cluster, was the most highly expressed miRNA in our ICM cells and is one of the major drivers of ground-state pluripotency in mouse ESCs[22]. Perhaps, there is a switch in the functional role of C14MC that is dependent on the window of development and a ramping up of expression is observed in the trophoblast post-implantation. Of note, although highly ICM-enriched, the average C14MC miR expression was 19-fold lower compared to the miRNAs of the C19MC (Supplementary Data 3).

C19MC is a maternally imprinted locus, which is upregulated in trophoblast and cell line derivatives, ESCs and germ cell maintenance[11,28]. In our TE cells, 72% (65/90) of the marker miRNA originated from the C19MC, all of which are located on the sense strand. Members of the C19MC were expressed at all developmental time points and lineages, though more highly expressed (22%) in the TE upon formation of the lineages, as evidenced in our enrichment analysis and TE-normalised ICM expression (Fig. 3d, Supplementary Data 3). We also observed that the magnitude of expression in the TE compared to ICM continues to diverge with time. Here, we demonstrate enrichment of C19MC in the preimplantation TE lineage, suggesting an early expression and perhaps a role in the regulation of trophoblast differentiation (Fig. 3d). Further studies are warranted to improve our understanding of the roles these clusters play in pre-implantation development.

## Non-canonical sncRNAs in preimplantation development

Next, we examined whether we could identify cell type-specific signatures for the other classes of sncRNAs. We observed minimal cell clustering when analysing expression of tRNA fragments, piRNA or snRNA by embryonic day or lineage (Supplementary Fig. 4a). This suggests that these biotypes undergo cell type-independent changes in expression level during the transition from 8-cell to differentiated blastocyst. In contrast, we were able to obtain distinct cell type signatures for snoRNAs expression (Fig. 3c, Supplementary Fig. 4a). SnoRNAs *SNORD115-116* from the paternally expressed long non-coding RNA host gene *SNHG14* were enriched at E3-E4, prior to lineage specification, while the snoRNAs from the paternally imprinted *MEG8* gene, *SNORD112-114*, were enriched in the ICM (Fig. 3c, Supplementary Fig. 4b,c). Although limited data implicate snoRNAs in lineage formation, some previous data suggest a dynamic profile in human ESCs[29] and during the morula to blastocyst transition in the bovine[30]. Interestingly, spliceosomal repression can effectively reprogramme mouse ESCs to a totipotent state, and *SNORD116* has unique effects on the spliceosome[31]. Previous profiling of 14 different tissues and cell lines suggest *SNORD114* has the highest expression in the ovary and is predicted to interact with and post-transcriptionally modify 18S rRNA[32]. The genomic co-localization of *SNORD112-114* and the ICM-enriched miRNA suggest that C14MC may be generating both snoRNA and miRNA precursors upon emergence of the ICM lineage. This is the first report of C14MC expression in the ICM, pointing to possible important role(s) of this locus in the maintenance of pluripotency and ICM lineage formation.

We next investigated tRNA expression dynamics as they have been previously shown to be developmentally regulated[29,30]. Although no obvious clustering by embryonic day or lineage was observed with UMAP analysis of aggregated tRNA count data (Supplementary Fig. 4a), we further separated tRNAs into 5′ halves (5′-tRFs), 3′ halves (3′-tRFs), and full length sequences and observed an increase in

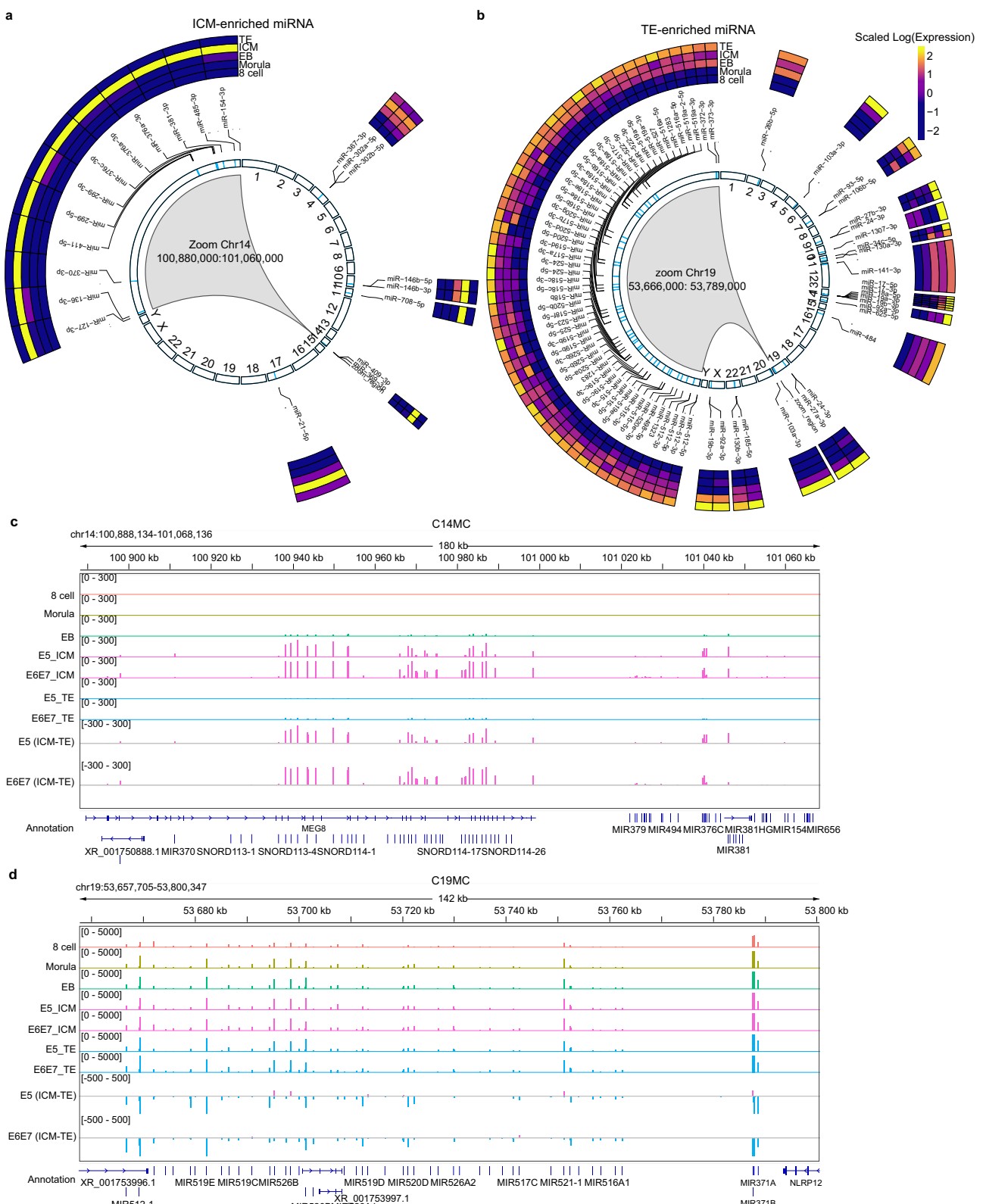

**Fig. 3 | Enrichment and expression of the C19MC and C14MC in early embryos.**
**a** Circular expression plot of ICM-enriched miRNA and their genomic origin, with expanded visualisation of C14MC, and similarly for (**b**) TE-enriched miRNA. IGV expression plots of relative sncRNA expression in our dataset from the (**c**) C14MC and (**d**) C19MC loci between lineages. Additional tracks subtracting the TE from ICM expression on E5 and E6/7 demonstrates enrichment of C19 in TE, over basal expression levels in the ICM. ICM inner cell mass, TE trophectoderm, E3-E7 embryonic days 3–7, EB early blastocyst.

proportion of full length tRNAs from E3 to E7 with a corresponding decrease in 5'-tRFs and 3'-tRFs (Supplementary Fig. 5a and b). Surprisingly, the overall distribution of Arginine and Asparagine codon usage represented by 3'-tRFs drastically decreased and increased over developmental time, respectively (Supplementary Fig. 5c). A similar expression pattern of 5'-tRFs were observed, in which Asparagine codons increased and Glutamyl codons decreased across development. The fragment data is in contrast to the codon usage in full length tRNAs which appeared consistent from E3 to E7, and in other codons which have limited differences in 5'-tRFs and 3'-tRFs (Supplementary Fig. 5c). While little is known about the roles of tRNA fragments in development, tRF[Arg] is able to donate its arginyl to Arginyltransferase, a highly conserved enzyme that causes mid-gestational embryonic lethality when knocked out in mice[33,34]. New mechanisms of gene regulation and post-translational modifications by tRFs and their binding partners are emerging, and additional work is needed to delineate their function in preimplantation embryogenesis (reviewed in ref. 7).

### Trajectory and pseudo-time analysis of miRNAs throughout preimplantation

Next, we sought to evaluate the changes in miRNA expression at a higher resolution throughout E3 to E7 development. The largest changes in miRNA differential expression (DE) were observed between 8-cell to morula (96 DE miRNAs) and morula to EB (163 DE miRNAs), while the initial formation of the ICM and TE was accompanied by modest changes (5 and 27 significantly differentially expressed miRNAs respectively, with log2 fold change more than 0.1 and fdr less than 0.05; Fig. 4a; Supplementary Data 4). During the maturation of TE from E5 (early blastocyst) to E6/7 (late blastocyst), we observed 74 DE miRNA (Fig. 4a), which included upregulation of the majority of the C19MC cluster miRNAs (e.g. miR-522-3p, miR-516b-5p, miR-512-3p) and downregulation of many pluripotency-related miRNA (e.g. miR-372-5p, miR-373-5p, miR-302b/d-5p). In contrast, no DE miRNA were observed in the ICM between E5 to E6/7, suggesting that once expressed, ICM associated miRNAs are maintained. These data demonstrate the divergence in miRNA expression patterns whereby in the TE, increased expression of trophectoderm-related miRNA markers are observed while concordantly decreasing their pluripotency, meanwhile ICM cells maintain expression of a set of miRNA during lineage maturation to E6/7.

We then inferred the pseudotime of our cells to address how miRNA expression patterns dynamically progress during human embryo development. We were particularly interested in identifying miRNAs which could be potentially driving lineage specification or maintenance. We observed a clear trajectory from 8-cell to EB, at which point two branches formed, one for the ICM and one for the TE (Fig. 4b). Generally, miRNA expression dynamics across pseudotime could be grouped into three categories. 1. Highly expressed in the pre-lineage, with a decrease in expression across time, irrespective of lineage. 2. Drastic increase in expression following ICM formation and 3. An enrichment in TE following specification. Several miRNA that were found to be highly abundant in the pre-lineages included miR-184, miR-204-5p, miR-508-3p, and miR-513a-5p (Fig. 4c). Given the relative quiescence of the human genome prior to embryonic genome activation (EGA), these miRNAs may be maternally inherited. For example, miR-184 is known to be required for drosophila germline development[35], and has been shown to be present in human ova along with miR-204-5p and miR-508-3p[16,36]. In contrast, development of the ICM trajectory is associated with an increase in several key miRNA (Fig. 4d). In mouse embryonic stem cells, miR-182 is required for self-renewal and maintenance of pluripotency[37] and has been shown to be among the top 20 miRNA secreted from human blastocysts, independent of lineage[38]. miR-376 cluster members, miR-376a-3p and miR-376c-3p, are significantly increased along the ICM trajectory, and have

previously been shown to target *IGFR1*, a key regulator of trophectoderm development[39,40]. Along the trophoblast trajectory, miR-27b-3p shows an exceptional rise, possibly participating in trophoblast differentiation through the suppression of pluripotency markers (Fig. 4e)[41]. Similarly, miR-519c-3p and miR-1323 are key members of C19MC which are highly expressed in trophoblast lineages[42].

With the abundance of C19MC and C14MC members among DE and trajectory-associated miRNAs, we aggregated expression across the entire cluster to better understand temporal changes in expression (Supplementary Fig. 6). miRNAs from the C14MC are relatively low until the EB window, during which their expression increases dramatically along the ICM trajectory but not along that of the TE (Supplementary Fig. 6a). Conversely, C19MC expression is already present at the 8-cell stage, rising to EB, and then diverges with higher expression in the TE vs ICM trajectory. These patterns are supported in ESCs, in which C14MC expression is drastically, and C19MC expression modestly, higher in naïve ESCs (Supplementary Fig. 6b)[29]. These data support a model in which the C14MC locus expression is activated in ICM, conferring a high-level of pluripotency, while expression of the C19MC locus is relatively high in both lineages but increases in the TE and decreases in the ICM with differentiation.

Furthermore, to verify the embryo and lineage specific expression of selected miRNAs using a different approach, we performed RT-qPCR on embryos biopsies (Supplementary Fig. 7). While expression of all miRNAs was confirmed in the human embryo, only miRNAs determined to be significantly enriched in the ICM were enriched in the RT-qPCR results (miR-146b-5p and miR-182-5p). RT-qPCR lacked the sensitivity to validate miRNAs determined to be significantly enriched in the TE by Small-seq, possibly due to the low number of embryos. This is likely due to the minimal difference in expression of 'TE enriched' miRNAs compared to those identified in the ICM (Supplementary Fig. 7).

### Developmental and lineage-specific miRNA targeting analysis

Canonically, miRNAs exert their influence through binding to the RNA-induced silencing complex (RISC), targeting the 3'-untranslated region of target gene RNA through base-pairing complementarity, and then either preventing translation (partial complementarity) or by endonucleolytic cleavage[43]. Next, we wanted to determine how developmentally-regulated miRNA could direct differentiation. We clustered our miRNAs into the 3 common expression dynamics across the ICM or TE trajectories, starting at 8-cell: C1=Low → High → Low, C2=Low → High, C3=High → Low (Fig. 5a, b). We restricted our targeting analysis to miRNAs following the C2 and C3 patterns, as we reasoned that they are the most likely to exert consistent directional effects on differentiation. By filtering for target genes with significant negative correlations with previous embryonic gene expression data[17], we identified 646 and 1098 miRNA-target genes in ICM, and 1230 and 1031 in TE from the C2 and C3 cluster types, respectively (Fig. 5c–f, Supplementary Data 5). We were specifically interested in miRNA-target-gene pairs that overlapped in both cluster direction and trajectory, such that the expression patterns of the miR-target pair were similar during progression from 8-cell to establishment of ICM or TE. This resulted in 1898 miR-target pairs common between the TE and ICM trajectories in the C2 pattern, and 1845 in the C3 pattern (Fig. 5g). Several developmentally-related predicted interactions were observed including the targeting and potential downregulation of the totipotency-related, zygotic genome activation (ZGA) markers *PRAMEF1* by miR-367-3p, *DPPA3* by miR-18a-5a, and *SLC34A2* by miR-373-5p in both trajectories[17,44]. These data further support the known role for a miRNA-mediated loss of totipotency during the transition from 8-cell to blastocyst. In zebrafish and xenopus embryos, zygotic miRNAs have been shown to participate first in translation repression followed by degradation of maternal RNAs during the ZGA[45,46], in contrast to mouse models where miRNA function is dispensable prior to the

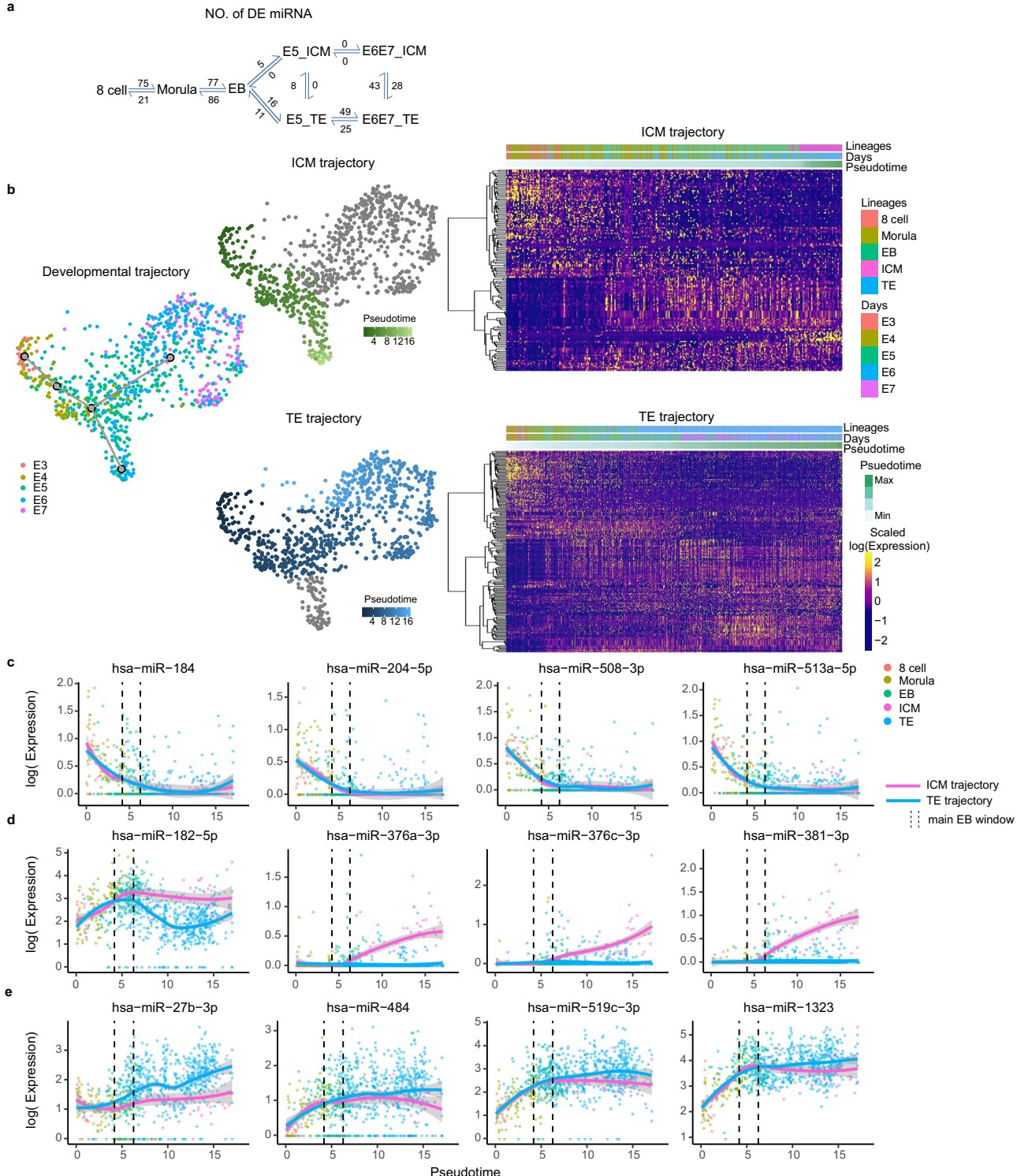

**Fig. 4 | Pseudotime analysis of miRNA expression in early embryonic development. a** Differential expression between developmental stages and lineages. **b** Developmental- and lineage-specific pseudotime trajectories with heatmaps of lineage trajectory-associated miRNA expression. Expression of several key lineage-associated miRNAs across pseudotime for (**c**) pre specification, (**d**) ICM, and (**e**) TE lineages. ICM: inner cell mass; TE trophectoderm, E3-E7 embryonic days 3–7, EB early blastocyst. The confidence interval (error bands, 95%) is indicated by bandwidth. The measure of centre and confidence intervals were calculated using the "loess" function with default parameters in R software.

blastocyst stage[47]. Our analysis revealed 188 genes potentially targeted and therefore downregulated by miR-373-5p, which follows a C2 pattern (steady increase) along both the ICM and TE trajectories (Supplementary Data 5). miR-373-5p, along with a host of other miRNAs which followed the C2 pattern (miR-302a/b-3p, miR-520b/c-3p, miR-

372-3p, miR-367-3p and miR-373-3p), may drive differentiation and an exit from 8-cell totipotency through coordinated targeting and downregulation of the early embryonic gene programme (Fig. 5h).

To determine the potential influence of miRNAs on lineage formation, we identified ICM- and TE-enriched miRNA signatures ($n = 31$

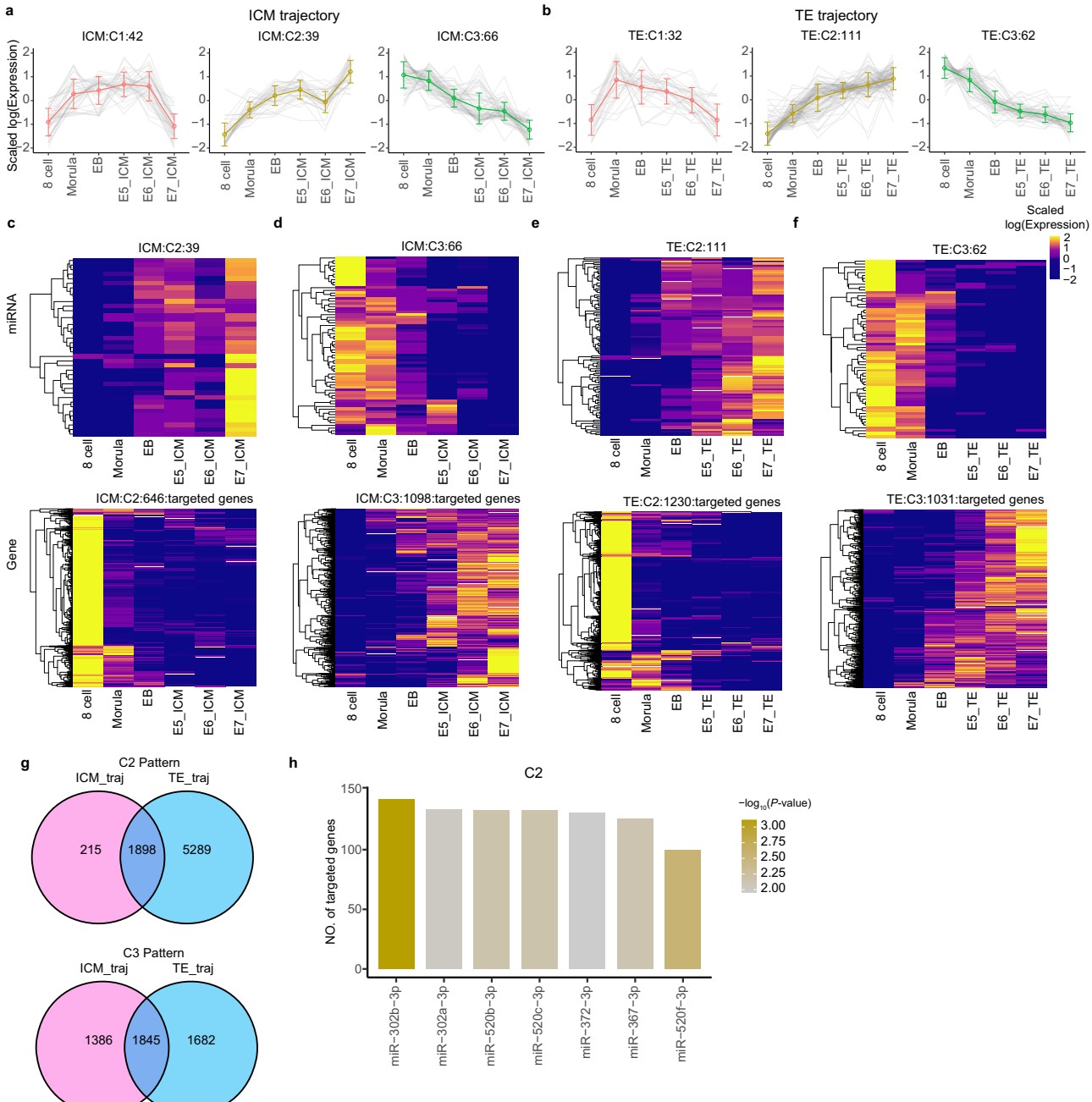

**Fig. 5 | Trajectory-related miRNA patterns and potential embryonic gene regulation.** K-means clustering of trajectory-related miRNA expression patterns in (**a**) ICM and (**b**) TE. C1=Low → High → Low, C2=Low → High, C3=High → Low. Error bars represent standard deviation values calculated among the miRNAs belonging to each cluster, the number of miRNAs of each cluster is indicated above. **c**–**f** Heatmaps of miRNA and target gene expressions in TE and ICM trajectories in C2 and C3 patterns. **g** Overlap between miRNA-target pairs between ICM and TE trajectories within C2 (top) and C3 (bottom) patterns of expression. **h** Number of target genes for C2 miRNAs with significant target enrichment over background. ICM inner cell mass, TE trophectoderm, E3-E7 embryonic days 3–7, EB early blastocyst.

and 19, respectively; Fig. 6a; Supplementary Data 6). Given the redundancy of miRNA targets, we 'pre-filtered' potential gene targets by focusing on the biologically relevant genes in human embryos[17]. As such, we leveraged our previously published single-cell RNA-seq dataset in the human and identified ICM- and TE- enriched genes ($n = 843$ and 659, respectively; Fig. 6b). Among the significant miRNAs, 21 and 14 were significantly negatively correlated ($p$ value < 0.05) with the gene targets identified ($n = 197$ and 71, respectively; Fig. 6c, d; Supplementary Data 6)[5]. Based on the repressive nature of miRNAs, we speculated that highly enriched miRNAs in the ICM may participate in lineage specification through targeting of TE-related genes, and vice

versa. Through screening of both predicted and validated gene targets, we generated miRNA-target networks for both lineages[48]. ICM-enriched miR-369-3p, miR-409-3p, miR-146b-3p, and miR-381-3p are predicted to target well known TE transcription factors, *GATA3, GATA2, TEAD1*, and *TET2* (Fig. 6e; Supplementary Data 6)[49–51]. Other well-characterised TE markers potentially targeted by enriched miR-409-3p, miR-374a-3p, and miR-378a-3p in the ICM include, *NR2F2, PEG10*, and *LRP2*, respectively[52,53]. Although none of these predicted miRNA-target interactions have been validated in embryos or hESCs, the 3'UTR of *NR2F2* has been shown to be targeted by miR-409-3p in both HEK293 and TZM-bl cell lines[54,55]. The enrichment of ICM miRNAs

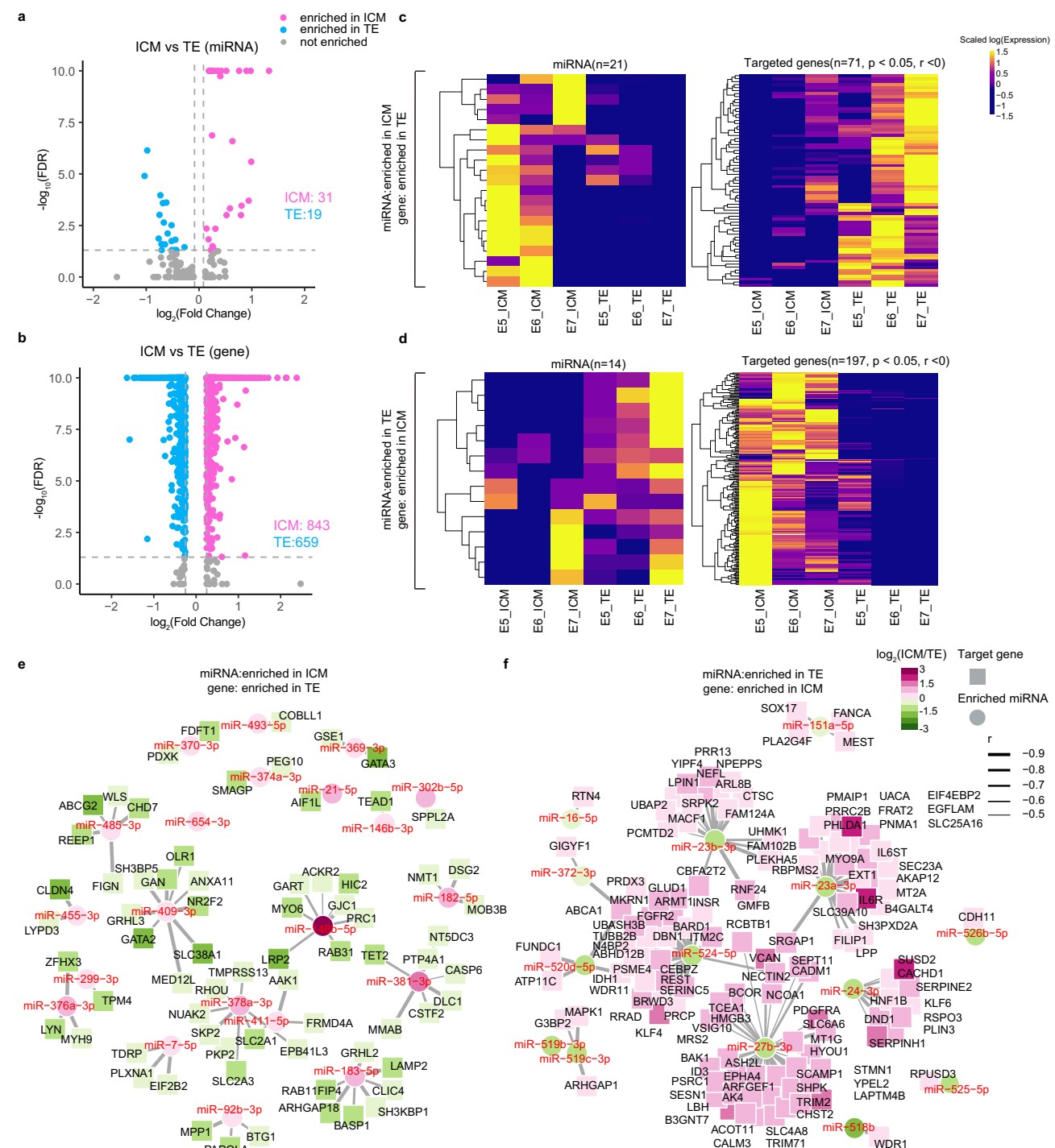

**Fig. 6 | miRNA targeting analysis and potential influence on lineage segregation. a** Volcano plots of miRNA and (**b**) their target genes in the ICM vs TE in E5-E7 cells. **c** Heatmaps of ICM-enriched miRNA expression (left) between lineages from E5-E7 and the corresponding TE enriched target genes (right). **d** Similarly for TE-enriched miRNA and their ICM-enriched target genes. *P* values of correlation were calculated by one-sided Student's *t* test. **e** miRNA-target gene network plot for ICM-enriched miRNAs and (**f**) TE-enriched miRNAs. For visualisation, the top 35 target genes of each miRNA (ordered by fold change) with at least 0.3 log2 Fold change are presented. Gene expression data reanalysed from[5]. ICM inner cell mass, TE trophectoderm, E3-E7 embryonic days 3–7, EB early blastocyst.

suggest a role in maintaining self-renewal and preventing differentiation to TE. Similarly, we observed TE-enriched miRNAs that have corresponding gene targets enriched in the ICM. These include miR-524-5p, targeting the well-known naïve pluripotency gene *KLF4*, and a particular enrichment of miRNAs targeting PE related genes such as miR27b-3p, miR-151a-5p, miR-524-5p, and miR-27b-3p which target *PDGFRA*, *SOX17*, *HNF1B*, and *IL6ST*, respectively[17,56–58] (Fig. 6f). Of these,

miR-151a-5p has been experimentally validated to reduce the expression of endoderm marker *SOX17* by luciferase reporter assays in cancer cell lines[59]. Further, ICM-enriched lysine demethylase *KDM7A* is targeted in the TE by four different TE-enriched miRNA: miR-27b-3p, miR-23b-3p, miR-23a-3p, and miR-524-5p. Zygotic knockdown of *KDM7A* in porcine embryos reduced blastocyst formation by 69%, dysregulated H3K9 and H3K27 methylation, and resulted in a reduced numbers of

ICM cells, indicating the importance of *KDM7A* on development of the ICM[60]. Collectively, these miRNAs may enforce TE identity.

Building on our findings of miRNA-mediated repression in lineage specification, we expanded our targeting analysis to explore the potential for miRNAs to also play a role in the upregulation of gene expression. Though we acknowledge that this phenomenon may be rare, studies suggest that miRNAs can also enhance gene expression under certain conditions[61,62]. With this in mind, we examined the possibility that lineage-specific miRNAs might not only suppress but also positively regulate target genes, thereby reinforcing lineage identity. Our analysis revealed 255 miRNA-target pairs within the ICM showing significant positive correlations, such as miR-493-5p with *SOX2*, miR-182-5p with *PDGFRA*, and miR-493-5p with *MAPK1* (Supplementary Data 6). In the TE, we identified 163 positively correlated pairs, including miR-27b-3p with *GATA3* and *GATA2*, as well as miR-524-5p with *NR2F2* (Supplementary Data 6). Therefore, it's possible that miRNAs play a dual role in regulating lineage-specific gene expression, highlighting pairs for future functional validation to elucidate their roles in reinforcing lineage specification or maintenance.

## Investigating miRNA function in mammalian preimplantation embryos

We next wanted to determine the functional role of miR-381-3p, as it was highly expressed in the ICM cells at the blastocyst stage (Figs. 2h, 4d), suggesting it may have an important function in ICM-TE specification or pluripotency. In accordance with this, miR-381-3p has been shown to be highly expressed and have important roles in both mouse and human naive pluripotency[23,63]. Given the limited access to 8-cell human embryos and the conserved expression of miR-381-3p in mouse and human[23,63], we chose to use the mouse model.

First, we confirmed that mouse embryos passively internalised miRNA mimics (Invitrogen™) (ThermoFisher Scientific), by spiking in a scrambled negative control mimic conjugated to Alexa Fluor 488, as others have also demonstrated[64] (Supplementary Fig. 8a). We next cultured mouse embryos with 10 μM miR-381-3p mimic, which prevented developmental progression to blastocyst stage, as expected given its enriched expression in the ICM and predicted targets related to TE formation (Fig. 7a). As we were interested in observing the role specifically in the ICM, we wanted to see if we could obtain some embryos that developed to the blastocyst stage by lowering the abundance of miRNA mimic. Indeed, with 5 μM miR-381-3p mimic, 45.8% of embryos were able to reach the blastocyst stage by E3.5 compared to controls (Fig. 7a).

To obtain a comprehensive overview of the impact of miR-381-3p mimic treatment on mRNA content and gene expression, we sequenced the transcriptomes of mouse embryos at the 8-cell and 32-cell stages from both control and miR-381-3p mimic-treated groups for 10 μM and 5 μM, respectively. After dimensional reduction using the top 2000 variable genes, we successfully identified the lineages for mouse embryo cells, confirmed by the expression of lineage markers such as *Spp1*, *Upp1*, *Cdx2*, and *Dppa1* (Supplementary Fig. 8d, e). Although cells from control and treated groups were clustered together in UMAP space (calculated using the top 25 principal components), we found that PC_2 and PC_3 represent gene expression variance from lineages and treatment, respectively (Fig. 7b). Further, we observed that ICM and TE lineage-specific genes were less distinguishable in treated groups (Fig. 7c, d), suggesting that the miR-381-3p mimic (5 μM) was modifying the transcriptional profiles of cells at the 32-cell blastocyst stage to resemble an earlier developmental time point. Considering the generally lower expression of ICM and TE related genes in 32-cell embryos, we then performed GSEA for the DEGs between ICM and TE in 32-cell embryos to more broadly examine the role of miR-381-3p at this specific stage (Fig. 7e and Supplementary Data 7). We observed that the expression level of genes enriched in the ICM (determined by comparing ICM vs TE control), were decreased in

the ICM cells derived from miR-381-3p treated embryos. In the TE cells derived from miR-381-3p treated embryos, the expression level of genes enriched in the TE (determined by comparing ICM vs TE control) decreased while the ICM enriched genes increased (Fig. 7e). This suggests that miR-381-3p is modifying the global gene signatures of the embryonic cells in both lineages making them less distinguishable. The regulation direction of DEGs in treated TE cells aligned with miR-381's role as an ICM-specific miRNA. However, it is puzzling to observe that DEGs highly expressed in ICM lineage were also underexpressed in the miR-381-3p mimic-treated group.

To examine the regulatory role of miR-381-3p at the protein level, including the potential impact on cell lineage allocation and the expression of ICM-TE lineage markers, we analysed surviving blastocysts treated with the 5 μM dose of the mimic using immunofluorescence. Treatment with 5 μM miR-381-3p mimic did not shift the proportion of ICM (Sox2+) and TE (Cdx2+) cells (Fig. 7f) nor the ICM:TE Ratio (Supplementary Fig. 8c) compared to controls. However, the expression of the lineage markers Sox2, Cdx2, and the computationally predicted target using TargetScan and miRDB, Tead1, significantly decreased in mimic-treated blastocysts (Fig. 7g, h and Supplementary Data 7). Notably, this was not due to a decrease in the total cell count of these early blastocysts (Supplementary Fig. 8b). Given that Sox2 and Cdx2 protein expression is lower in mouse embryos at earlier developmental timepoints, this suggests that although these embryos have reached cavitation and formed blastocysts, they are more representative of earlier timepoints in terms of molecular expression, in corroboration with our scRNA-seq analysis above. Together, when considering both the low and high dose of miR-381-3p mimic, our experiments suggest that this miRNA may play an important role in maintenance of ICM-like phenotype and/or prevent the TE programme.

## Novel embryonic miRNAs

Given that sncRNA expression in human preimplantation is relatively unexplored, we speculated that our dataset may provide insight into the presence of novel, highly-context specific miRNA. We identified six novel miRNAs, two of which passed strict abundance, sequence, and loci characteristics utilising miRDeep2 (Supplementary Data 8)[65]. These novel miRNA share a seed sequence with miRNAs from mouse and *C. elegans*[66], suggesting they are evolutionarily conserved. Furthermore, these two miRNA candidates met additional criteria for annotating new miRNA[67], and have been reported previously[16,68], providing further confidence in our detection. The predicted structures and expression levels are presented in Supplementary Figs. 9–13.

The previous identification of novel_miRNA_4_chr4_29031 in pre-EGA human oocytes and embryos[16], and our observation that it decreases from 8-cell to blastocyst (Supplementary Fig. 9), suggests that it may be maternally inherited. Sequence abundance and structural analysis predicted the 3' end of this novel miRNA to be the mature form (Supplementary Fig. 9). Next, we applied the same method as with the DE miRNA targeting and identified potential gene targets of this novel miRNA which display significantly inverse expression patterns (Supplementary Data 8). Significantly downregulated targets of novel_miRNA_4_chr4_29031 included *KMT2A*, a histone methyltransferase and *SMARCD2*, a chromatin remodeler. Previous knockdown experiments of *KMT2A* by miRNA targeting in mouse zygotes reduced H3K4 trimethylation and blastocyst development[69], leading to the hypothesis that the observed decrease in novel_miRNA_4_chr4_29031 from E3 to E7 may permit translation of *KMT2A*. Similarly, SMARCD2 is a chromatin remodeler that is upregulated during preimplantation development[70]. We now speculate that novel_miRNA_4_chr4_29031 contributes to epigenetic reprogramming from zygote to blastocyst. Future functional studies of these novel miRNAs and their embryonic gene targets are warranted.

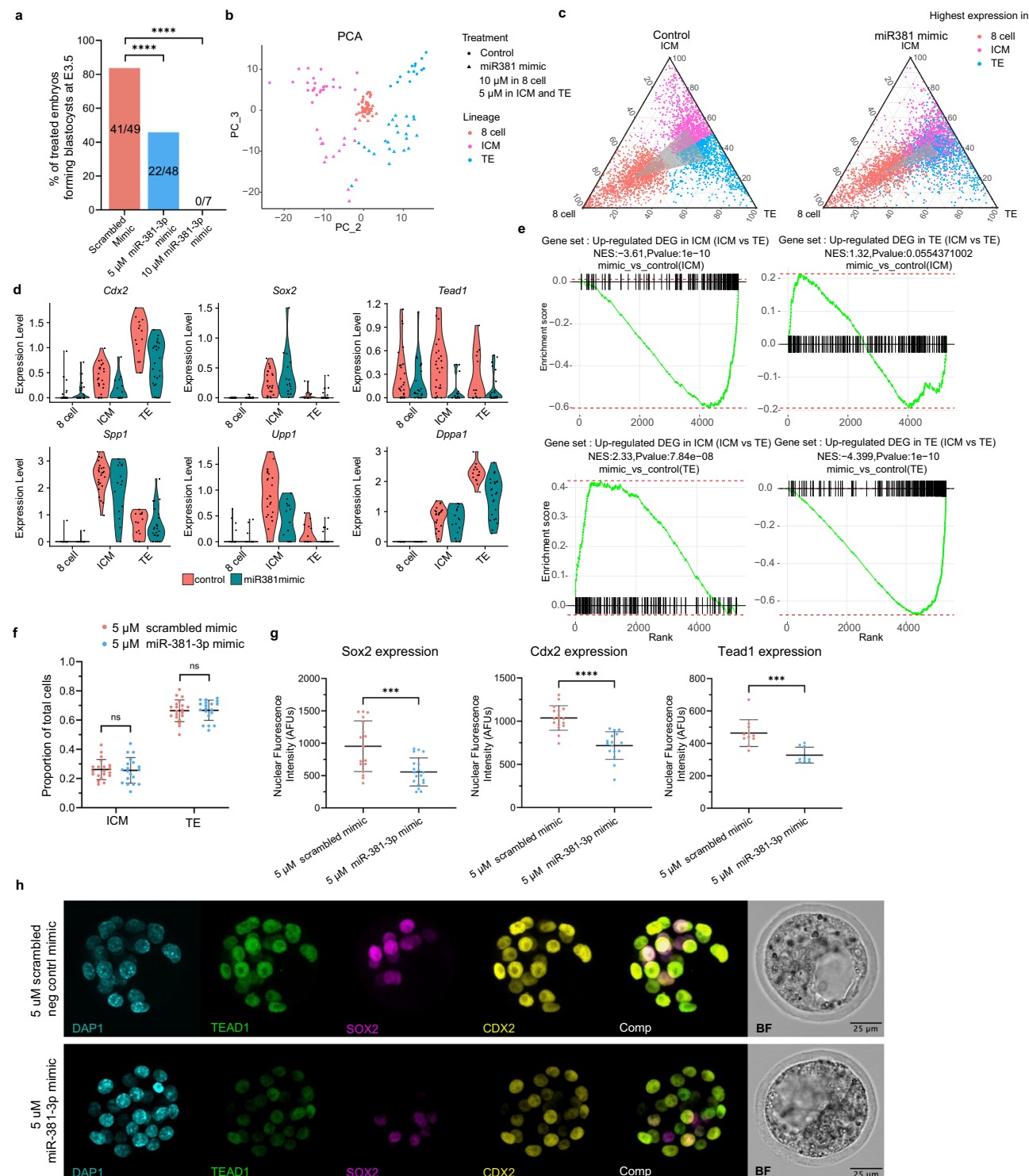

## miRNA isoform analysis

Several studies have described the presence of miRNA variations in length and/or sequence due to physiological events occurring in vivo, termed isomiRs (reviewed in[71]). These variations include elongations/trimming of either end of the mature sequence and non-templated (post-transcriptional) additions to the 3' end[72]. Therefore, we asked if embryonic cells exhibited any consistent modifications across development or within specific isomiRs. First, we observed that the overall proportion of miRNAs without length variations was 39.2% at E3, consistently rising to 43.5% at E7 (Supplementary Fig. 14a). The most abundant length variations included 3'

elongations, 3' trimming, and 5' trimming. The 21 isomiRs with the highest abundance were clustered by expression from E3 to E7 (Supplementary Fig. 14b). Several of the isomiRs displayed interesting dynamics across development. The 3' trimmed isoform of miR-23a-3p is highly expressed at E3, but decreased drastically across development, meanwhile its 3' elongated isoform consistently increases starting at E5. miR-23a-3p has been shown to display tissue-specific isoform expression and strong evolutionary selection[73], while isomiRs as a whole have distinct regulatory activities from their canonical counterparts[74]. The dynamic regulation of miR-23a-3p isomiRs, among others, points to potential developmental-specific

**Fig. 7 | The role of miR-381-3p during mouse preimplantation development.**
**a** Percentage of total embryos treated with scrambled negative control or miR-381-3p mimic that developed into blastocysts at embryonic day (E)3.5. Statistical analysis was conducted using the Chi-Square Test. Values on the bar graphs correspond to: n embryos that developed to blastocyst/total n embryos in the group.
**b** Principal component analysis (PCA) plot based on scRNA-seq gene expression data from mouse embryos in control and miR-381-3p mimic-treated cells, where colour represents cell identity and shape represents the different treatments.
**c** Ternary plot showing the average expression of DEGs in 8-cell, inner cell mass (ICM), and trophectoderm (TE) cells in control and mimic-treated embryos. DEGs were detected in control embryonic 8-cell, ICM, and TE cells, and further classified by the lineage with the highest gene expression. The vertices of the grey triangle inside the plot indicate the mean expression proportion in the three lineages for each type of DEG. **d** Violin plot shown selective marker expression in control and mimic-treated embryos. **e** Gene set enrichment analysis (GSEA) plot showing enrichment scores for genes with at least a 0.1 log2(fold change) difference related to DEGs determined between ICM vs. TE control, mimic-treated TE vs. control TE, and mimic-treated ICM vs. control ICM. *P* values from the one-sided Kolmogorov–Smirnov test are indicated in the figure heading. **f** Analysis of cell lineage allocation in early (32-cell) blastocysts treated with scrambled control versus 5 μM miR-381-3p mimic, where each data point represents the proportion of total embryonic cells belonging to the ICM or TE in a single embryo (*n* = 20 control embryos, pink and *n* = 19 mimic embryos, blue). Lineage was assigned based on the exclusive expression of known lineage markers Sox2 (ICM) and Cdx2 (TE) using immunofluorescence and confocal microscopy. Statistical analysis was conducted using a two-sided Student's t-test, where values are reported as means ± standard deviation. **g** Average nuclear fluorescence intensity (protein expression, measured in arbitrary fluorescence units (AFUs)) of Sox2 (in ICM cells), Cdx2 (in TE cells), and Tead1 (in all cells) in early blastocysts treated with scrambled control versus 5 μM miR-381-3p mimic. Each data point represents the average intensity of the cells in one embryo. Statistical analysis was conducted using the Mann-Whitney or a two sided Student's t test where appropriate. Values are reported as means ± standard deviation. **h** Representative immunofluorescence and brightfield (BF) images of control and mimic-treated blastocysts analysed in (**f**, **g**). Staining/imaging was performed in 5 separate batches on *n* = 11–17 control and *n* = 9–19 mimic-treated embryos. DAPI = nuclear stain. Comp = composite image. BF Brightfield. *ns* = *p* > 0.05, *** = *p* < 0.001, **** = *p* < 0.0001.

roles which are adjacent or complementary to their canonical sequences.

We also asked if isomiRs with non-templated additions (NTAs) were present and variable in our dataset. NTAs were less abundant than length variations (average of 13.9% of UMI), but still followed a decreasing trend from E3 to E7 (Supplementary Fig. 14c). The most common NTA was the addition of one to three 3′ adenosines, which has been shown to have a stabilising or destabilising effect on miRNA, depending on the cellular context[9,72]. In our data, 8 of the top 10 isomiRs with the NTA of adenosine decreased from E3 to E7 (Supplementary Fig. 14d). In contrast, two of the most influential miRNA based on our trajectory analysis, miR-367-3p and miR-302c-5p, were the two miRNA with an increasing abundance of 3′ adenosine additions from E3 to E7. Our current knowledge pertaining to the role of isomiRs during embryo development, irrespective of species, is limited and warrants further exploration.

## Discussion

Our study illuminates a previously unexplored aspect of early human development. Here, we investigated the developmental dynamics of sncRNA expression from E3-E7 of human embryo development. We profiled and quantified all major sncRNA biotypes, including miRNA, piRNA, snoRNA, tRNA and rRNA fragments at single-cell resolution, taking into account developmental time and lineage. Of these, miRNA and snoRNA appear developmentally regulated in the E3 to E7 window of compaction and blastulation, suggesting important roles of these biotypes in potency, lineage specification and/or maintenance. In particular, miRNA clustering and targeting analyses demonstrate the intricate role of miRNAs in the transition from totipotency to pluripotency within the two distinct ICM and TE lineages. These data provide a rich resource for developmental and stem cell biologists which can be leveraged for comparison to other model systems and functional experiments to validate the regulatory roles of sncRNAs in embryogenesis.

Studies of human sncRNA dynamics in the embryo are very limited. To date, only one study has profiled sncRNAs spanning from human oocytes until 8-cell embryos at single-cell resolution, and found peak expression of piRNAs in human oocytes and zygotes which gradually decreased to the 8-cell, while miRNAs generally increased with development[16]. Further, we observed that the miR-371-373 cluster, also identified prior to EGA and in bulk, probe-based analysis of human blastocyst miRNAs, continues to increase in abundance from E3 to E7 along both lineages, albeit higher in ICM, congruent with its close relationship to establishment of naive ESC pluripotency[29,75]. This cluster is one of the main loci activated with the primed-to-naive ESC conversion, and syntenic with the mouse miR-290-295 cluster which is highly expressed in naive mouse ESCs and the preimplantation ICM lineage (reviewed in ref. 27). Previous studies have implicated a host of miRNA in the process of human ESC and iPSC establishment and differentiation - two systems which closely model the epigenetic and transcriptional signatures in preimplantation embryos (reviewed in ref. 27). Naïve ESCs have been shown to reflect the preimplantation EPI, with the capacity to generate all somatic lineages and the germline. The seminal papers on miRNA differences between primed and naïve ESCs demonstrated a robust increase in the expression of the miR-371-373 cluster[17], similar to what we observed from E3 to E7 in both lineages, with the highest expression in ICM cells. Specifically, we identified 188 targets of miR-373-5p which were downregulated in the same developmental window (including *PTEN*, *LATS1*, and *USP9X*), suggesting that this may be a hub miRNA mediating the transition into pluripotency. The miR-302/367 family also follows an increasing trend with development, with over 400 of their targets decreasing in expression from E3 to E7. Transfection of a few miRNA including the miR-302/367 cluster in both human and mouse somatic cells induced dedifferentiation into iPSCs[76]. Mechanistic studies have determined that the miR-302/367 cluster is involved in regulating the cell cycle and mesenchymal-to-epithelial transition in both iPSCs and ESCs[77]. Expression of miR-302/367 strengthens TGFβ signalling through targeting of *LEFTY1/2* transcripts, and elevates bone morphogenic protein (BMP) signalling by targeting the transcripts of *TPB2*, *DAZAP2*, and *SLAIN1*, consequently enhancing TE fate[78,79]. The functions of the remaining host of miRNA which were enriched from E3 to E7 remain to be determined.

Our study shows miRNAs can be overexpressed in preimplantation embryos simply by spiking mimics in the culture media, and that these miRNAs can have an impact on blastocyst formation and lineage specification. Treatment of mouse embryos with a mimic for miR-381-3p resulted in lower blastocyst rates and the decreased expression of lineage specific genes. It is worth noting that a recent study which combined Small-seq with single-cell mRNA sequencing to give a more detailed look into miRNA functions in mouse embryonic stem cells (mESCs), found that compared to DROSHA knockout cells (which are unable to synthesise miRNAs canonically), wildtype cells only saw an -15% repression of computationally predicted TargetScan genes of the most highly expressed miRNAs[80]. This suggests that the regulatory role of miRNAs is a complex phenomenon, which may be attributed to the necessity of large miRNA clusters in order to observe drastic changes in gene expression, as may be the case in important biological systems, such as establishment of lineage or maintenance pluripotency, and/or that miRNAs may have important functions that are not always

associated with repression. Further experiments will be required in order to determine the exact mechanisms by which specific miRNAs act in order to elucidate their functional roles during preimplantation development. It is important to note that Tead1 has been shown to be critical in the specification of the human TE[51], whereas its role in the mouse TE is minimal[81], suggesting the possibility of different outcomes upon over-expressing miR-381-3p in human embryos. Furthermore, a cross-species comparison will determine the degree to which results from mouse studies, such as these, can be extrapolated to human preimplantation development, hESCs and the primate in general.

Our cluster analysis shows a distinct population of TE cells, which increases proportionally by E7. Ninety miR were enriched in the TE cells, with 72% originating from a single locus, the C19MC miRNA cluster. C19MC expression maintains the epithelial cytotrophoblast stem cell population of the placenta via the suppression of the epithelial-to-mesenchymal transition[82]. Although expression of C19MC cluster members were enriched in TE, they also appeared at the 8-cell stage with pervasive expression across lineages and developmental time, suggesting a possible role in the maintenance of pluripotency in the embryo. In support of this, C19MC activity was recently shown to be a hallmark of naïve human ESCs, and disruptions in C19MC, either through knockout or epigenetic silencing, hindered the conversion of primed ESCs to a naïve state[11]. Further, reactivation of C19MC in primed ESCs facilitated the generation of TE stem cells. Perhaps the observation of C19MC in both the ICM and TE might reflect the ICM's plasticity and inherent capacity to contribute to the TE lineage. Knockout of C19MC in embryos will help to elucidate the specific roles of C19MC in TE development within a three-dimensional context and in maintaining pluripotency. However, the exclusivity of C19MC to primates, coupled with the ethical constraints on genetic manipulation of human embryos, necessitates these studies be conducted in non-human primates or stem-cell derived models.

Conversely, C14MC, a conserved miRNA cluster in eutherian mammals, expression was observed only at the EB stage, just prior to ICM-TE specification and remained exclusively expressed in the ICM. Though the function of C14MC remains poorly understood, the kinetics observed here suggest a specific role in ICM-TE divergence. Furthermore, a recent report demonstrated the importance of three C14MC members, miR-541-5p, miR-410-3p, and miR-381-3p, in the maintenance of pluripotent ESCs and upregulation of *Oct4*, *Nanog*, and *Klf5*[22]. Therefore, it is possible that C14MC miRNAs, among the other ICM-specific miRNAs identified, act to maintain ground-state pluripotency in human ICM post cavitation and support the ICM-TE lineage formation.

The major factor limiting the efficacy of Assisted Reproductive Technologies (ARTs) is failed implantation following embryo transfer. From the embryo-perspective, it is difficult to discriminate between embryos which are implantation competent versus those which will fail to implant. Given their importance in cell-cell communication and the process of implantation, many groups have aimed at identifying miR-NAs derived from spent blastocyst culture media as noninvasive markers to predict successful pregnancies[25,38]. However, one major limitation in using miRNAs captured in the spent culture media, is that the embryo origin could not be traced as until now, lineage specific miRNA profiles had not been established at this resolution. The data contained herein provide a catalogue of sncRNAs which can be exploited for a lineage-of-origin based analysis of spent blastocyst media which will provide insight into the pathology of the embryo and the source lineage of potentially dysregulated sncRNAs. These findings have the potential to refine IVF culturing conditions and pave the way for targeted interventions, like miRNA mimics, to bolster embryo quality and competence. As the trend continues to move towards single embryo transfer, our sncRNA atlas offers a promising source of non-invasive biomarkers to enhance the selection of competent blastocysts.

Our study comprehensively details the landscape of sncRNAs, at single cell level, throughout human preimplantation development, including a nuanced analysis of their temporal and lineage-specific dynamics. The expression patterns of miRNAs and snoRNAs reveal an intricate relationship with cellular differentiation. The marked decline in piRNAs and tRNAs, set against an increasing trend in miRNAs, snoRNAs, and rRNA fragments, underscores the changing sncRNA milieu as the embryo progresses from E3 to E7. This study also sheds light on novel sncRNAs, contributing additional layers to the already complex coordination of molecular architecture during preimplantation development.

Furthermore, our identification of cell type-specific miRNA signatures, especially the prominence of the miR-376 and C14MC clusters in ICM cells and the C19MC cluster in TE cells, provides new insights into the possible mechanisms driving lineage specification and maintenance. The discovery of genomic hotspots linked to these sncRNA profiles emphasises the potential of these loci in orchestrating the developmental potential. Clinically, our findings support the role of miRNAs as essential arbiters of cell metabolism, signalling, and differentiation, offering a valuable repository for elucidating the impact of sncRNA dysregulation in clinical conditions like embryonic arrest, irregular blastocyst formation, and implantation failure. Together, this comprehensive analysis enhances our understanding of early embryogenesis and opens avenues for targeted therapeutic interventions in regenerative medicine and reproductive health. Further, this atlas may be leveraged to assist with benchmarking stem cell-based embryo models and naive human embryonic stem cell lines to their in vivo counterparts.

## Methods

### Ethics approval and consent to participate

All supernumerary embryos used for this study were from couples attending the CReATe Fertility Centre between 2002 and 2007 who have completed family building and were electing to terminate storage of their embryos. At that time, patients were presented with the option of discarding their embryos or donating to research with stated goals including "improving our understanding of embryo development and treatments for infertility". Informed consent was obtained from those electing to donate to research with a form which was approved under the University of Toronto REB protocol #30251, Veritas IRB protocol number #16580, Stockholm Regional Ethics Board (4 -1255/2018) and the Université de Montréal and Centre de Recherche du Centre Hospitalier de l'Université de Montréal regional ethics board (CERSES-20-107-R and 20.126, respectively). Participants were not compensated for their embryos as per the Canadian Assisted Human Reproduction Act.

### Standards involving human participants

All procedures involving human participants in this study were conducted in compliance with the ethical standards outlined in the Tri-Council Policy Statement: Ethical Conduct for Research Involving Humans (TCPS 2) and the Declaration of Helsinki.

### Human embryo culture and dissociation

Supernumerary human embryos donated to research by patients attending CReATe Fertility Centre, with study approval by the University of Toronto (protocol # 30251), Veritas IRB (protocol number 16580), Stockholm Regional Ethics Board (2018/691-31) and the Université de Montréal and Centre de Recherche du Centre Hospitalier de l'Université de Montréal regional ethics board (CERSES-20-107-R and 20.126, respectively) were utilised. Embryos were thawed using the Vit Kit-Thaw (FujiFilm). A total of 87 embryos were cultured from embryonic days 3–7 in a SAGE 1-step medium. Embryos were then dissociated into single cells as previously described using TrypLE enzyme solution and micromanipulation with glass capillaries[17,42,83]. For a proportion of embryos, ICM and TE enriched was performed by

biopsy in the clinic, to ensure adequate acquisition of all cell types. Cells were manually collected into individual 8-strip PCR tubes containing 2 µL of lysis buffer (0.13% Triton X-100, 1.33 U/µL Recombinant RNase inhibitor, 2.0 µM 5.8S rRNA-masking oligonucleotide) and stored at −80 °C until further processing[15]. Embryo sex was not considered in this analysis since sex determination based on small ncRNA expression is not feasible. Of note, we did not observe any segregation of cells within lineage or developmental time to suggest a stratification of data by sex. As such, for all downstream experiments, cells from male and female preimplantation embryos were grouped.

### Co-seq to capture sncRNA and mRNA from a single-cell

To identify embryonic lineages based on transcriptomic signatures, single-cell mRNA-sncRNA Co-seq was performed as previously described and validated on mouse embryos[14]. Briefly, individual cells were collected into 4 µL of lysis buffer. After cell lysis at 72 °C for 20 min, lysates were split into two equal 2 µL halves, with one half allocated to sncRNA sequencing[14,84] and the other to Smart-seq2[5,14,84]. Each half was prepared following the original Small-seq and Smart-seq2 protocols, using 22 amplification cycles during the first PCR of Smart-seq2. This approach enabled the association of mRNA markers of cell lineage with corresponding sncRNA profiles.

### Pre-processing and quality control of mRNA portion of Co-seq data

For single-cell Co-seq data, read quality was firstly checked by FastQC (v0.11.9)[85]. Subsequently, we aligned high-quality reads to the human GRCh38 reference genome (v.3.0.0, GRCh38, obtained from the 10X Genomics website) using the STAR aligner (v2.5.1b) with default settings[86]. Raw read counts were quantified using the rsem-calculate-expression from RSEM (v1.3.0)[87] with the option of " --single-cell-prior". To filter low quality cells, a cut-off based on the number of expressed genes (nGene) of 750 and percentage of mitochondrial genes (percent.mito) of 0.3 was used, leaving 103 cells post-filtering. Gene expression was normalised based on the library size excluding mitochondrial genes.

### Integration of mRNA portion of Co-seq data with embryonic reference data

To determine the cell-types within our Co-seq dataset, we integrated the the Smart-seq2 (mRNA) portion with our previous scRNA dataset[5] using the canonical correlation analysis (CCA) approach implemented in the R package Seurat (v4.2.0)[88] based on 15 dimensions and 2000 anchor features. After integration, Principal component analysis (PCA) was performed on the scaled data from the integrated object followed by embedding into low dimensional space with Uniform Manifold Approximation and Projection (UMAP) as implemented by the "RunUMAP" function. For clustering, the shared Nearest Neighbour (SNN) Graph was constructed on the PCA embedding by calling the "FindNeighbors" function followed by the identification of clusters using the "FindClusters" function. The assignment of cell identities to the formed clusters was accomplished by leveraging well known gene signatures for lineages[17,42,89].

### Pre-processing and quality control of Co-seq sncRNA data

SncRNA reads were processed using a modified version of the Small-Seq pipeline[29]. First, the UMIs (8 bp associated with the unique molecular identifiers) were removed from sequence reads and appended to the reads' name using umi_tools (v0.4.4)[90]. Then, cutadapt(v1.17)[91] was used to remove adapter and polyA sequences (parameters: -e 0.1 -o 3 -m 18 -M 81 -u 2). The setting of "-M 81" was chosen because our small RNA sequencing had a maximum read length of 101 base pairs. Additionally, at least 10-base pair trimming of adapter or polyA sequences was performed to ensure that the reads originated from short transcripts. The resulting clean reads were aligned to GRCh38-3.0.0 using

bowtie (v1.0.0) with the following parameters: "-a --best --strata -v 2 -m 50 -S -q -p 2"[92]. To avoid false positive mapping, reads with a length less than 20 base pairs and with one or more mismatches were excluded and reads with lengths between 20 and 40 base pairs and two mismatches were also filtered out. Mismatches located at the last base pair of reads were disregarded, considering the "CCA" addition in tRNA maturation and miRNA modification dynamics. In addition, unmapped reads underwent recursive soft clipping by 1 nucleotide from the 3' end and were then mapped to the reference genome using Bowtie with parameters "-a --best --strata -v 1 -m 50" until the soft-clipped read length reached 3 nucleotides, the same procedure for 5' end of reads was subsequently performed. Any reads with a mapping length of less than 17 base pairs were discarded. PCR amplicons were collapsed, and RNA molecules were counted using UMI-tools with the "dedup --method directional" option. Deduplicated soft-clipped reads were remapped onto the genome using the same procedure as above to recover multiple mapping information dampened by the "dedup" process. Annotation coordinates on the genome were then extracted from Mirbase V22 (miRNA)[66], Gencode V39 (snoRNA, snRNA, rRNA, and others)[19], GtRNAdb V19 (tRNA)[93], piBase V3.0[94] to assign the reads to their corresponding categories. The reads were assigned to each category in a hierarchical manner based on their overlap with the above annotations in the same strand. If a read was mapped to more than one category, we assigned it to only one category based on the following order of priority: miRNA, rRNA, snoRNA, snRNA, tRNA, piRNA, and other types of transcripts. Reads overlapped with mature miRNA and piRNA were further required to be less than 40nt. Subsequently, cell quality filtering was implemented based on three criteria: First, each cell required a minimum of 0.5 million sequenced reads. Second, the proportion of mitochondrial associated UMIs was required to be less than 25%. Third, the number of expressed miRNA molecules per cell was required to be more than 100. Data generated using Co-seq were only used for data integration and were excluded from downstream analysis to minimise introduction of batch differences from library preparation and as such introduce technical errors.

Primate-specific miRNAs were extracted from Hu et al.[95] by requiring human miRNAs with orthologs in chimpanzee, gorilla, orangutan, macaque, and marmoset at the same time, but not in mouse, rat, dog, cow, opossum, or chicken. Primate non-specific miRNAs were defined as those having orthologs in mouse, rat, dog, cow, opossum, or chicken. miRNAs that did not meet these conditions were classified as "uncertain".

### miRNA expression analyses for Co-seq data

miRNA from E6 and E7 embryos were selected if they exhibited one or more counts in at least 2 cells. Log-normalised counts were calculated using the deconvolution strategy implemented by the "compute-SumFactors" function in scran package (v.1.14.6)[96]. Dimensional reduction analysis was performed using the Seurat package. The top 150 most variable miRNAs were identified by "FindVariableFeatures" function using the "vst" method followed by regressing out variance caused by the number of expressed miRNA by "ScaleData" function. Subsequently, the top 15 principal components were calculated using the "RunPCA" function and utilised for UMAP dimensionality reduction and clustering using "RunUMAP" and "FindClusters" function respectively. To label clusters based on miRNA expression, we leveraged indicator cells from the scRNA-Seq portion of Co-seq and assigned them as either "ICM" or "TE". Marker miRNAs were pinpointed through the use of the "FindAllMarkers" function.

### Small-seq library preparation and sequencing

The remaining, non-Co-seq sncRNA libraries were generated using the Small-seq protocol as previously described[15]. Briefly, cells were thawed and incubated at 72 °C for 20 min followed by 3' adapter ligation at 30 °C for 6 h followed by 4 °C for 10 h. Free adapters were then

removed by 5' deadenylation followed by digestion of RT primer-3' adapter duplexes by lambda exonuclease. The 5' adapter was ligated at 37 °C for 1 h, followed by reverse transcription with Superscript II (ThermoFisher Scientific) at 42 °C for 1 h. Two PCRs were performed, first to amplify the cDNA products (12 cycles) and second to barcode the libraries for multiplexing (12 cycles). The resulting pooled libraries (~150 nt) were processed with the DNA Clean & Concentrator (Zymo) at a 7x buffer:DNA ratio, eluted in 50 μL, analysed by 2100 Bioanalyzer (Agilent) and quantified by Qubit DNA High Sensitivity (ThermoFisher Scientific). Libraries were sequenced on either NovaSeq or NextSeq 550 100 bp SE or PE for a sequencing depth of 2 million reads per cell. During the following analysis, only the forward read was used from PE data.

### Pre-processing, quality control and normalisation of Small-Seq data

Read mapping, feature counting and normalisation for each batch of full-cell Small-seq data were using the same procedures as those employed for the Small-seq portion of the Co-seq data. After getting the Log-normalised counts, we performed rescaled normalisation using the "multiBatchNorm" function in the batchelor package (v.1.2.4)[97] so that the size factors were comparable across batches. The log-normalised expression after this rescaling step was utilised in marker gene detection and differential gene expression analysis. naïve, Primed and HEK cells from[29] were processed using the same pipeline by setting the length cutoff for 30 bp.

### Integrated analysis of small-seq datasets

We integrated a total of five different batches, with four originating from full-cell Small-seq and one from Co-Seq. The integration was achieved through the CCA approach from the Seurat(v4.2.0) package, employing 15 dimensions and 150 anchor features, with a "k.filter" setting of 50. Integration was conducted separately for miRNA, tRNA, piRNA, snRNA, and snoRNA expression datasets, with the exclusion of sncRNAs only linked to the sex chromosomes. Following integration, PCA was applied to the scaled data from the integrated object, followed by embedding into a low-dimensional UMAP space and clustering. Above steps were all performed by utilising Seurat(v4.2.0) package. The final assignment of cell identity for the formed clusters was determined by leveraging the miRNA signatures obtained from the Co-seq dataset with known lineages.

### Marker miRNA detection and differential miRNA expression analysis

Final miRNA Markers for each clusters were identified using "FindAllMarkers" function from the Seurat(v4.2.0) package. Top markers with p_val_adj <0.001 were selected (Supplementary Data 2). Visualisation of markers miRNA loci were performed using R package circlize(v0.4.12)[98]. Normalisation size factor for merged reads used in IGV[99] were calculated by DESeq(v1.38.0)[100]. Differential miRNA expression analysis between different lineages as well as different developmental time points was performed using the two-sided 'wilcox' test implemented in "FindMarkers" function and FDR were calculated using "Benjamini-Hochberg" method. miRNAs with an FDR less than 0.05, log2 (fold change) greater than 0.1, and expression in more than 33% of the cells were considered as differentially expressed. Differential gene expression analysis between ICM and TE cells were performed using the same function with log2 (fold change) greater than 0.25 and FDR less than 0.05 for cells from[17].

### RT-qPCR analysis of miRNA in embryo biopsies

To facilitate ultra-low input miRNA RT-qPCR, a universal poly(A)-based RT method coupled with miRNA-specific DNA priming was employed as previously described[101]. Biopsies from 8 day 6 donated blastocysts were collected by a trained embryologist first from the ICM and then the TE. The two biopsies were transferred to 1 μL of lysis buffer (see above), and stored at −80 °C until RT. Upon thawing, lysates were incubated at 72 °C for 3 min and transferred directly to ice. All reagents are given at final concentrations. Poly(A) tailing was performed in a final volume of 2 μL with 1x PAP buffer (New England Bioabs (NEB) Cat#M0276), 1 mM ATP (NEB Cat#M0276), and 0.25 U/μL Poly(A) Polymerase (NEB Cat#M0276) at 37 °C for 30 min, then placed on ice. Prior to RT, 1 μM RT primer (IDT; see sequence in Supplementary Table 1) and 0.5 mM dNTPs (NEB Cat#N0447S) were added to a final volume of 3 μL, and the reaction incubated at 65 °C for 5 min then placed on ice. Next, 1x M-MuLV buffer, 40 U/μL M-MuLV RT (NEB Cat#M0253), and 2250 U/μL RNAse Inhibitor (Takara Cat#2313B) were added to 4 μL, and incubated at 42 °C for 1 h, then at 95 °C for 5 min, then cDNA was stored at −20 °C until used for qPCR. To measure the relative abundance of lineage-enriched miRNA, cDNA was diluted from 4 μL to 40 μL and 2 μL was added to each 10 μL qPCR reaction containing 1x PowerTrack SYBR Green Master Mix (ThermoFisher Cat#A46012) and 400 nM forward and reverse primers (IDT; Supplementary Table 1), with cycle conditions following manufacturers recommendations. Reactions were performed in triplicate. miR-26a-5p was selected as an internal reference gene, as it had low variability between cells and was not significantly different between ICM and TE. Naive induced pluripotent stem cell RNA from line SCTi003A (STEMCELL Technologies) was used as a reference sample. Data were analysed with the 2^-ddCt method[102] including a student's t-test to compare expression between TE and ICM.

### Pseudotime reconstruction and trajectory inference

Single-cell pseudotime trajectory based on miRNA expression was computed using the R package slingshot (v.1.8.0), which enables computation of lineage structures in a low-dimensional space[103]. In brief, pre-computed cell embeddings and clusters obtained from the Seurat integration served as the input of the function "slingshot" by setting the start cluster to "E3", followed by function "slingPseudotime" to infer individual pseudo time with setting "na=T". Because individual pseudotime were calculated separately for ICM and TE trajectory, we needed to align the pseudotime between these two trajectories. This alignment was achieved through the following steps:

$$\text{scale.factor} = (\text{Psd\_TE\_max} - \text{Psd\_prelineages\_max})/$$
$$(\text{Psd\_ICM\_max} - \text{Psd\_prelineages\_max})$$

$$\text{Psd\_ICM\_mod} = (\text{Psd\_ICM} - \text{Psd\_prelineages\_max}) * \text{scale.factor}$$
$$+ \text{Psd\_prelineages\_max}$$

Here, Psd_TE_max represents the latest pseudotime for TE cells, Psd_ICM_max represents the latest pseudotime for ICM cells, and Psd_prelineages_max represents the latest pseudotime for pre-lineages cells. Psd_ICM represents the individual pseudotime for ICM cells, which is greater than Psd_prelineages_max, and Psd_ICM_mod represents the rescaled individual pseudotime for ICM cells.

To visualise individual pseudotime, we employed principal curves on the integration object, excluding Co-Seq Small-seq datasets and "unclassified clusters". We identified miRNAs associated with the trajectories using the "testPseudotime" function from the R package TSCAN (v1.28.0)[104] by blocked influence from sequenced batches with FDR less than 0.05. Additionally, we further selected those by requiring them to be differentially expressed through comparisons of miRNA expression profiles at two adjacent stages or comparisons of expression profiles between each later time point with E3. These analyses were stratified by the two trajectories (ICM and TE) separately. Expression patterns of the trajectory-related miRNA were further explored using the "k-means" method using R package pheatmap(v1.0.12)[105] by setting k = 3.

## Analysis of mRNA gene targets from identified miRNA

Candidate miRNA-mRNA target relationships were collected from mirDB[48] and TargetScan V7.2[106] (union set). To determine the miRNA-mRNA target relationship related with ICM and TE lineage segregation, only differentially expressed miRNA and mRNA were utilised. Pearson correlation coefficient between pseudo bulk log-transformed miRNA expression and targeted gene expression were calculated for ICM and TE at E5, E6, E7. One-sided Student's t-tests were performed to test for negative or positive correlation, considering that mRNA and miRNA expression are typically inversely correlated, only targets with negative significant correlations were shown in the figures and all significant correlations were included in the supplementary data. For miRNA targeting genes during ICM trajectory, miRNA and targeted genes were filtered to be ICM trajectory-related miRNAs and differentially expressed genes among different timepoints. Pearson correlation coefficients between pseudo bulk log-transformed miRNA expression and targeted gene expression were calculated at the level of 8-cell, morula, EB_pre-lineage, E5 ICM, E6 ICM and E7 ICM. Similar calculations were performed for the TE trajectory.

## Identification of novel miRNAs

Putative novel mature miRNAs, which were not previously annotated in miRbase, were detected using mirDeep2 (v2.0.1.2)[65]. To increase the identification power, deduplicated reads that did not overlap with known non-coding RNAs (ncRNAs) were collected and collapsed together, subsequently evaluated by each sequenced batch and developmental timepoints. We employed the mirDeep2 tools, utilising the "mapper.pl -r 20" and "miRDeep2.pl" wrapped scripts for mapping and detecting novel miRNAs. Default parameters were used without further specification. Candidate novel miRNAs meeting the following criteria were selected. 1. Identified as "significant" according to "significant_randfold", 2. possessing a positive miRDeep score, 3. Being expressed in at least two of the batches or developmental timepoints, and 4. having a minimum length supported by at least 30 cells, with 75% of overlapped reads falling into the 20–25 nucleotide range. To predict the targeted mRNA of these novel miRNAs, we utilised miRDB (https://mirdb.org/cgi-bin/custom.cgi) by uploading the sequences of the mature novel miRNAs[107]. The Pearson correlation coefficient between pseudo-bulk, log-transformed novel miRNA expression and the expression of their predicted target genes was calculated at 8-cell, morula, EB, E5-E7 TE and ICM.

## Collection, treatment, and culture of mouse preimplantation embryos with miRNA mimics

All mouse experiments were approved by the Centre de Recherche du Centre Hospitalier de l'Universite de Montreal (CRCHUM) Comité Institutionnel de Protection des Animaux du CHUM (CIPA) under the protocol number IP21005SPs. Mice were maintained in individually ventilated cages (up to 5 animals/cage) at $22 \pm 2\,°C$ and 40-60% humidity in 12 h light/dark cycle with lights on from 6:30am-6:30 pm with ad libitum access to food and water. To induce ovulation, C57BL/6NCrl (Charles River Laboratories) x DBA/2NHsd (Inotiv/Envigo) F1 females (6-12 weeks old) were injected with 5 IU of pregnant mare's serum gonadotropin (PMSG), followed by 5 IU of human chorionic gonadotropin (hCG) 48 h later. Super-ovulated females were placed with C57BL/6NCrl x DBA/2NHsd F1 male mice for mating immediately following the hCG injection and sacrificed using isoflurane anaesthetic and cervical dislocation 45–48 h later. At this time (E1.5), uteri were dissected and flushed with 37 °C EmbryoMax M2 medium (Sigma-Aldrich, cat # MR-015-D) as previously described[83] to collect 2-cell embryos. These embryos were placed in pre-equilibrated EmbryoMax KSOM Media (Sigma-Aldrich, cat # MR-121-D) droplets covered with EmbryoMax Filtered Light Mineral Oil (Sigma-Aldrich, cat # ES-005-C).

mirVana™ custom miRNA mimics (Invitrogen™ (ThermoFisher Scientific)) of miR-381-3p (5 μM or 10 μM) or a scrambled negative control mimic (Invitrogen™ (ThermoFisher Scientific), cat # 4464058) at the same concentration were added directly to media droplets containing 2-cell embryos. Alternatively, some 2-cell mouse embryos were collected and treated as described above but with a custom version of the scrambled negative control mimic conjugated to Alexa Fluor 488 (Invitrogen™ (ThermoFisher Scientific)) to verify mimic uptake. Embryos were cultured with their respective treatments to later stages of preimplantation development in a normoxic incubator at 37 °C (5% $CO_2$, 20% $O_2$).

## Single cell RNA sequencing and data processing of mouse pre-implantation embryos

Following culture, mouse embryos were collected for single cell dissociation, isolation, and library preparation using Smart-seq2, as previously described[17,83,108] at either the 8-cell (E2.5) or 32-cell (E3.5) stage, for both control and mimic-treated embryos (3–5 embryos/group/timepoint)[17,83,108]. Sample purification, downstream library preparation, sequencing with a NextSeq550 and data analysis using in-house pipelines were all performed as previously described[17,108].

The sequencing reads of the Smart-seq2 data from the mouse dataset were processed using the same pipeline as for the mRNA portion of Co-seq data, but mapped onto the mouse mm10 reference genome (mm10-2020-A, obtained from the 10X Genomics website). Low-quality cells were filtered out by discarding those with fewer than 3000 expressed genes or with a percentage of mitochondrial genes higher than 0.1. Following normalisation, based on library size excluding mitochondrial genes, the standard Seurat pipeline was followed. Specifically, genes expressed in at least 3 cells were log-normalised for downstream analysis. The 2000 most variable genes were identified using the "vst" method with the "FindVariableFeatures" function. The top 25 principal components were then calculated using the "RunPCA" function and utilised for UMAP dimensional reduction with "RunUMAP" function. Clusters were identified using the "FindClusters" function with resolution set to 0.6. Differentially expressed genes (DEGs) were identified using the two-sided 'wilcox' test implemented in the"FindMarkers" function, and FDR were calculated using the "Benjamini-Hochberg" method. Genes with an FDR less than 0.05, a log2(fold change) greater than 0.25, and expression in more than 33% of the cells were considered differentially expressed. Candidate miRNA-mRNA mouse target relationships were collected from mirDB and TargetScan (union set). Gene set enrichment analysis was implemented using the "fgsea" function from the R package fgsea (v.1.28.0)[109].

## Confocal microscopy and immunofluorescence analysis of mouse embryos

Control and mimic-treated embryos were collected at the 32-cell blastocyst stage (E3.5) and fixed (in 4% PFA (Electron Microscopy Services, cat # 15714) for 20 min at RT), permeabilized (in 0.3% Triton X-100 (Sigma-Aldrich, cat # X100-500ML) for 15 min at RT), and blocked (in 3% BSA (Wisent, cat # 800-095-EG) for 3 h at RT) for analysis via immunofluorescence and one photon confocal microscopy. Embryos were incubated overnight at 4 °C with the following primary antibodies: anti-Sox2 (eBioscience™, cat # 14981182, lot # 2493179, 1:100), anti-Cdx2 (BioGenex, cat # MU392A-5UC, lot # MU392A0820, 1:250), and anti-Tead1 (Cell Signalling, cat # 12292, lot # 4, 1:100). Subsequently, embryos were incubated with the following secondary antibodies for 2 h at RT: Alexa Fluor donkey anti-rabbit 488 (Life Technologies, cat # A21206, 1:1000), Alexa Fluor donkey anti-rat 555 (Life Technologies, cat # A48270, 1:1000); and Alexa Fluor donkey anti-mouse 647 (Life Technologies, cat # A31571, 1:1000). Finally, embryos were placed in a 1:10000 dilution of Hoechst nuclear stain (Invitrogen, cat # H1399) for 10 min at RT.

All images were acquired using an Olympus FV1000 upright microscope (Olympus, Japan), equipped with a XLUM Plan FL N 20x/1.00 Water objective. Embryos were positioned in PBS between two

coverslips (#1.5, 0.18 mm) with spacers (SecureSeal™,Grace Bio-Labs). For excitation, 405 nm (solid state), 488 nm (Argon laser), 543 nm (solid state) and 635 nm (solid state) lasers and a A DM405/488/543/ 635 dichroic mirror were used. For detection, photomultiplier tubes (PMT) detectors were set as follows: a first SDM490 was positioned in front of the first PMT associated with a BA 430-470 for DAPI detection; then a SDM560 was positioned in front of the second PMT associated with a BA 535-565 for AF488 detection; a SDM640 was positioned in front of the third PMT associated with a BA 560-660 for AF594 detection; and finally, a mirror was positioned in front of the last detector with a BA 655-755 for AF647 detection. All images were acquired sequentially (frame mode) as follows: AF488 and AF647 simultaneously first, AF555 second, and DAPI last. Laser settings (power, HV, offset) were kept consistent between all images and experiments. Images were acquired using the Olympus Fluoview software (v4.2.4.5, Olympus, Japan) in a $512 \times 512$ pixel format at zoom 4 (pixel final resolution of 310 nm), 4μs/pixel speed, and with a line Kalmann of 3. Z-stacks were acquired with a 1 μm step size.

Images were analysed using ImageJ software version 2.14.0/1.54 f to assess blastocyst formation, lineage cell counts, and to quantify the relative protein expression of genes of interest. Nuclear fluorescence intensity (arbitrary fluorescence units) measurements from each cell in all channels were normalised to the average expression of Hoechst (nuclear stain) within the corresponding embryo. Statistical analysis was performed using GraphPad Prism 4 statistical software. All values were reported as means ± standard deviation. Differences between control and treated groups were analysed using the Chi-square, Mann-Whitney, or two sided Student's t test where appropriate. $P < 0.05$ was considered a statistically significant difference.

### Characterization of sncRNA modifications
We conducted the above recursive mapping strategy to study the 5' and 3' tailing of sncRNAs while minimising error in false positive mapping. In comparison to the mature miRNA sequences listed in miRbase, we identified isomiRNAs by annotating miRNA sequences exhibiting a shift of −3 to +3 nucleotides and the presence of non-templated nucleotides at the 3' ends. For tRNAs, those with read lengths shorter than 50 nucleotides and a shift of at least −10 nucleotides at the 3' end (but not at the 5' end) relative to the reported loci in GtRNAdb were categorised as 5' tRNA halves. Conversely, tRNAs displaying a similar shifting pattern at the 5' end were classified as 3' tRNA halves. Those tRNAs with lengths exceeding 50 nucleotides and demonstrating no such shifts more than −10 nucleotides at both ends were designated as full-length tRNAs.

### Reporting summary
Further information on research design is available in the Nature Portfolio Reporting Summary linked to this article.

## Data availability
The data supporting the findings of this study are available from the corresponding authors upon request. Raw read sequencing files (FASTQ), along with unfiltered read count expression matrices, have been deposited in the Gene Expression Omnibus (GEO) database with accession number GSE249713. The expression matrices for the pre-implantation embryo dataset used in this study were sourced from ArrayExpress, accession number E-MTAB-3929. Source data for all graphs/figures are provided as a Source Data File. Source data are provided with this paper.

## Code availability
Original code has been deposited on Zenodo under https://zenodo.org/records/12818575 for all versions. Any additional information required to reanalyse the data reported in this work paper is available from the lead contact upon request.

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

## Acknowledgements

The computations and data handling were enabled by resources in project NAISS 2023/22-988 and NAISS 2023/23-490 provided by the National Academic Infrastructure for Supercomputing in Sweden (NAISS), partially funded by the Swedish Research Council through grant agreement no. 2022-06725. This work was supported by funding from the Swedish Research Council (2016-01919, SP), Swedish Society for Medical Research (S16-0039, SP), The Canadian Institutes of Health Research (PJT-178082, SP). SP holds the Canada Research Chair in Functional Genomics of Reproduction and Development (950-233204, SP). Additional funding was provided internally by the CReATe Fertility Centre. We'd like to acknowledge Siamak Bashar for setting up the culture of research-donated embryos and Matthew Hamilton for his assistance with library preparation. Dr. William Pastor and Jessica Cinkornpumin (McGill University) for assistance with revisions. The CRCHUM Cellular Imaging Core Facility for their equipment and expertise with imaging procedures. Finally, the authors thank the patients from CReATe Fertility Centre for the donation of embryos.

## Author contributions

S.P. conceived and supervised the study. C.Z. performed the bioinformatic analysis. S.J.R., S.B., K.M., C.Z., M.H.–J. and S.P. performed the experiments, analysis and interpretation of data. S.P., S.J.R., S.B., C.Z., M.H.–J. and C.L.L. contributed to the writing of the manuscript.

## Funding

## Competing interests

The authors declare no competing interests.
