## [Peer Review File · Nature Communications]

An atlas of small non-coding RNAs in Human Preimplantation DevelopmentREVIEWER COMMENTS

Reviewer #1 (Remarks to the Author):

The authors here present a systematic and careful profiling of small RNAs and mRNAs in early human development, covering single cells of the E3-7 developmental stages. The complementary application of Small-seq and Co-seq to single cells is really beyond-state-of-the-art. Further, the computational analyses are extensive and sound (but see below) and the biological findings are very well-integrated into existing literature. Overall, this is a very well-conducted study that will be useful to the non-coding RNA and developmental biology communities, both in terms of pushing biological insights, as a resource and as setting new standards for methods applications.

Major comments:

1. Overall the sequence data generation and computational analyses seem very sound. However, I have concerns related to the miRNA targeting inference. This is based on the union of mirDB and TargetScan, and the miRNA-target relations are further filtered by considering only cases where the expression of the miRNA and the putative target are negatively correlation across the early development stages. As a sanity check of this miRNA-target inference, it will be important to compare how many miRNA-target expression changes are concordant with target repression (negative correlations) as opposed to discordant (positive correlations) as this will give some idea of the false positive rate for the predicted miRNA-target relations. There are many reasons why expression of a given miRNA and transcript can change in different conditions (for instance regulation by opposed TFs) and direct miRNA repression is just one of those reasons.
2. The mirDB database for miRNA-target interactions is relatively widely cited, but has not like for instance TargetScan been extensively benchmarked against Omics data generated from miRNA perturbation studies. To my understanding, mirDB also bases its miRNA predictions heavily on CLIP-seq data, which is known to contain many peaks that reflect non-functional 'scanning-type' interactions by AGO (Agarwal, Bell, Nam and Bartel 2015). Therefore it would be interesting to know what the above analyses look like if only TargetScan (and not mirDB) is considered - and if there are fewer 'discordant' relations as described above.

Minor comments:

3. Given the abundances of different classes of small RNAs, it would be interesting to see if some classes might have been missed in the analyses. It would be interesting to see supplementary versions of Figure 1D and 1E that include non-annotated sequences. If these sequences are abundant and have specific length profiles, it could suggest that some classes (like siRNAs) have been overlooked.
4. The observation that small RNA abundances decrease during these early developmental stages, driven by a decrease in tRNA fragments and piRNAs in very interesting. What is the evidence that the huge abundance of small RNAs in the E3 stage is biological, rather than technical? Did the authors consider using spike-ins to normalize small RNA abundances?
5. The addition of novel miRNAs (Suppl. Fig. 6) adds even more value to the study. It would be useful

to include the miRDeep2 PDF output figures of read distributions as a new supplementary figure.

Reviewer #2 (Remarks to the Author):

The authors have presented a comprehensive profile of small non-coding RNAs (sncRNAs) in human preimplantation embryos. Within this, it was observed that miRNAs target known lineage-specific marker genes, showing a consistent pattern. Furthermore, correlations in expression were also demonstrated. The variations in the expression of modified tRNAs and isomiRs are novel and intriguing. Such a resource is unprecedented and represents highly significant information. It is suggested that for this study to be more suitable for publication in Nature Communications, it needs to go beyond just being a source of information and discussing correlations in expression changes. The study should also delve into deeper analysis.

Major Points:

1. The identification of C14MC as a hotspot in ICM-lineage expression is noted, but it would be beneficial to include information on the role it plays experimentally, similar to what has been done with C19MC in human ES cells.
2. Similarly, for the newly identified miRNAs by the authors, those that are adequately expressed in human ES cells should undergo functional validation. This would greatly enhance the quality of the paper.
3. The Discussion section currently feels redundant with a list of prior studies. It should be revised to include functional speculation and be more cohesive with the findings of the current study.

Minor Points:

1. Line 58: Correct "scnRNA" to "sncRNA."
2. Supplementary Figure 2B legend: Change "Morual" to "Morula."
3. In Figures 3C and 3D, the IGV graphs for E5 (ICM-TE) have a shorter y-axis compared to others. This should be adjusted to match the length of the other axes.
4. Line 196: Is this referring to Supplementary Figure 3?
5. Supplementary Figure 6: The depiction of miRNA_33190 needs revision.

Reviewer #3 (Remarks to the Author):

The paper Russell et al. provides a novel atlas of small non-coding RNAs in human preimplantation development.

The group generated a novel sncRNAs dataset along with an extensive bioinformatical analysis. They look into the abundance and kinetics of different type of sncRNAs across different stages of human embryonic day. They found sncRNAs corresponded to specific genomic hotspots, and enrichment of specific miRNAs clusters in different lineages (e.g. TE vs. ICM). They also identified six novel miRNAs, which seem to be conserved across species, and may have embryonic functions.

This dataset has the potential to be useful to the stem cell and developmental biologist community. However, I have two major concerns that I would like the authors to address. Firstly, the way the human embryos are staged is based on embryonic days. This has been shown to be flawed and it does not properly benchmark the embryos to the developmental time. In addition, I think it would add value to the paper to validate some of the identified sncRNAs with expression characterization and functional assays in human embryos (preferably) or other embryos, or in vitro models like human

embryonic stem cells or human trophoblast stem cells.

Major points:

- Embryonic stage evaluation of human embryos.

This paper stages the human embryos using embryonic days (E3 to E7). The authors use this method as this was used also in their previous paper (Petropolous et al., Cell, 2016). However, since then, several other works pointed out the importance to stage the embryos based on morphokinetic analysis rather than embryonic days (Gerri et al., Nature, 2020; Regin et al., Human reproduction, 2023; Zhu et al., Elife, 2021; Firmin et al., Seminars in Cell and Dev Bio, 2021; and others). What is E3/E4/E5/E6/E7 equivalent to? How do the embryos look like at E3-E7? 8-cell? Morula? Cavitating morula? Expanded blastocyst?

Using embryonic stages can work for species like the mouse, where pre-implantation embryos undergo a very stereotypic developmental progression, however frozen/thawed human embryos are very asynchronous. While there is no consensus in the field about the best way to stage the embryos, the authors should take inspiration from the papers mentioned above and make an effort to assess the human embryos morphologically before subjecting them to single cell dissociation. They should also make a scheme and show images of the representative embryos from E3 to E7, and describe in the methods how they do the staging of the embryos.

- Further characterization and validation of key identified sncRNAs.

As mentioned throughout the manuscript, validating the role of sncRNAs is possible in both embryos and stem cell lines. I would like to ask the authors to do some experiments to support the possible involvement of some of the key sncRNAs identified. Can the authors look at the expression of some preferentially expressed miRNAs in TE and ICM via staining (e.g. RNA scope, RNA-ISH?) in human embryos? Could they assess their potential role using knockdown approaches in human embryos? If human embryos are not possible for such experiments, could they validate this in other species, e.g. mouse, if they are conserved? Otherwise, could some functional test be done in hESCs or hTSCs?

Minor points:

- Stirparo et al. (Development, 2018) reassessed the single cell transcriptomic data developed in human embryos. One of the datasets that they considered in their reanalysis was Petropolous et al., Cell, 2016. They raised the concern of possible TE over-representation in the Petropolous et al. dataset, and suggested that possibly there was incomplete immunolysis and ICM recovery in the embryos used in this study. I am wondering whether the authors took into account the re-analysis of Stirparo et al. when using the Petropolous et al. dataset in the current work. More importantly, do the authors performed ICM immunosurgery in this paper as well? I would like to ask them to clearly state this in the method and elaborate about the protocol used as much as possible, and not only cite their previous references.

- I would like to ask if the authors could add the human zygote in their sample pool to have a complete view of the sncRNA dynamics.

- Is there some analysis that the authors could do to look into primate-specific sncRNAs?

Dear Reviewers,

We thank all reviewers for providing constructive feedback and comments. We have thoroughly revised our manuscript (NCOMMS-23-57491) and have addressed all comments.

Here are the point-by-point responses.

REVIEWER COMMENTS

Reviewer #1 (Remarks to the Author):

The authors here present a systematic and careful profiling of small RNAs and mRNAs in early human development, covering single cells of the E3-7 developmental stages. The complementary application of Small-seq and Co-seq to single cells is really beyond-state-of-the-art. Further, the computational analyses are extensive and sound (but see below) and the biological findings are very well-integrated into existing literature. Overall, this is a very well-conducted study that will be useful to the non-coding RNA and developmental biology communities, both in terms of pushing biological insights, as a resource and as setting new standards for methods applications.

Major comments:

1. Overall the sequence data generation and computational analyses seem very sound. However, I have concerns related to the miRNA targeting inference. This is based on the union of mirDB and TargetScan, and the miRNA-target relations are further filtered by considering only cases where the expression of the miRNA and the putative target are negatively correlation across the early development stages. As a sanity check of this miRNA-target inference, it will be important to compare how many miRNA-target expression changes are concordant with target repression (negative correlations) as opposed to discordant (positive correlations) as this will give some idea of the false positive rate for the predicted miRNA-target relations. There are many reasons why expression of a given miRNA and transcript can change in different conditions (for instance regulation by opposed TFs) and direct miRNA repression is just one of those reasons.

Response:

We appreciate the reviewer for raising this important point. We have included the positively correlated miRNA-gene pairs in our revised Supplementary Tables 5 and 6 and lines 306-318 and 354-358. Additionally, we have now examined the overall number of significantly correlated miRNA-gene pairs, stratified by the source of the database, which includes both positively and negatively correlated targeting pairs (Reviewer Figure 1A). Regarding the number of significantly correlated miRNA-gene pairs ($p < 0.05$), we did not observe a

dramatic difference in the proportion between positive and negative pairs based on whether the targeting relationships were recorded in miRDB or TargetScan only.

Reviewer Figure 1. A. Proportion of significantly correlated miRNA-gene pairs (P-value < 0.05) stratified by the source of the database. Bars above or below the x-axis indicate the proportion distribution with positive or negative values of the correlation coefficient, respectively. The correlation calculations were performed for scenarios including ICM vs. TE, associated with ICM trajectory, or TE trajectory. **B.** Histogram showing the general distribution of miRNA-gene pairs based on correlation coefficient values, stratified by the source of the database.

2. The mirDB database for miRNA-target interactions is relatively widely cited, but has not like for instance TargetScan been extensively benchmarked against Omics data generated from miRNA perturbation studies. To my understanding, mirDB also bases its miRNA predictions heavily on CLIP-seq data, which is known to contain many peaks that reflect non-functional 'scanning-type' interactions by AGO (Agarwal, Bell, Nam and Bartel 2015). Therefore it would be interesting to know what the above analyses look like if only TargetScan (and not mirDB) is considered - and if there are fewer 'discordant' relations as described above.

Response:

We thank the reviewer for raising this important point. The original version of miRDB was built on CLIP-seq data. However, the 2020 version which we have leveraged includes datasets and algorithms built on miRNA overexpression and RNA-seq datasets, providing much more accurate predictions^{1,2}. In addition, we checked the overall correlation coefficient distribution stratified by the source of the database (Reviewer Figure 1). We find that generally, there are not fewer proportions of discordant relations predicted whether miRDB or TargetScan analysis is performed.

Furthermore, we have now included functional validation of miR-381-3p, which is conserved between mouse and human³⁻⁵). Please see lines 367-418, methods section lines 786-859, revised Supplementary Data 7, Fig. 7 and Supplementary Fig. 8. Our data shows that when treating mouse embryos with miR-381-3p, followed by single cell RNA sequencing, we observe concordance with predicted targets for both human and mouse by miRDB and our data, whereby downregulation of *TET2*, *TRA2A* and *TEAD1* ($FDR=0.0542$) is observed at the mRNA level (please see revised Supplementary Data 7). Further, spiking miR-381-3p mimic in the culture media of mouse embryos at a dose of 10 μ M resulted in embryo arrest varying between the 4 cell to 16 cell stages, while at a dose of 5 μ M, around 50% developed into blastocysts (see revised Fig. 7a and Supplementary Fig. 8). Imaging of blastocysts exposed to 5 μ M did indeed demonstrate a significant decrease in TEAD1 protein (see revised Fig 7g-h).

Minor comments:

3. Given the abundances of different classes of small RNAs, it would be interesting to see if some classes might have been missed in the analyses. It would be interesting to see supplementary versions of Figure 1D and 1E that include non-annotated sequences. If these sequences are abundant and have specific length profiles, it could suggest that some classes (like siRNAs) have been overlooked.

Response:

We thank the reviewer for raising this point. We originally focused our analysis to include only the most common classes of small RNAs, particularly those that have previously been examined in preimplantation development using other methodologies. We have now included a revised Supplementary Fig.1, which depicts the genomic localization of the non-annotated sequences by overlapping existing annotations from GENCODE and Repbase. Small-seq utilises UMIs which are strand-specific and representative of the sense strand. In the current analysis, we are displaying the proportion of UMI in both the sense and antisense. Please see lines 121-128 in the revised manuscript. Length distribution of the most abundant ncRNA overlapped with genomic localization of lncRNA and LTR were checked in revised Supplementary Fig. 1. We are uncertain how to further profile these and relate them with a biological meaning in the embryo and as such have left this additional analysis as a supplement.

4. The observation that small RNA abundances decrease during these early developmental stages, driven by a decrease in tRNA fragments and piRNAs is very interesting. What is the evidence that the huge abundance of small RNAs in the E3 stage is biological, rather than technical? Did the authors consider using spike-ins to normalize small RNA abundances?

Response:

The usefulness of spike-ins is debated in the field of single cell genomics, as they carry their own innate biases, though recently a ‘gold standard’ has been developed for single cell RNA-sequencing to account for potential amplification biases⁶. For single cell small ncRNA-seq protocols, the majority, including the one utilized to generate this dataset, do not employ a spike-in strategy, but instead use Unique Molecular Identifiers⁷⁻⁹. Small-seq

is an UMI based method that circumvents the technical artefacts associated with amplification that is frequently observed with single-cell OMICs data, such as PCR duplicates. Further, we believe the distribution pattern observed is not technical for the following reasons: 1. Previous literature supports the inheritance of tRNAs and piRNAs from maternal oocyte and paternal sperm, which fits the progressive decrease in with development^{10,11}. This has also been previously demonstrated in a recent paper profiling small ncRNAs in human oocytes to 8 cell embryos¹². 2. The expression of PIWL machinery decreases with development (revised Fig. 1e), supporting the decrease in expression of piRNAs. 3. A similar trend would likely be observed for the other ncRNAs, such as miRNAs, if this decrease was technical.

5. The addition of novel miRNAs (Suppl. Fig. 6) adds even more value to the study. It would be useful to include the miRDeep2 PDF output figures of read distributions as a new supplementary figure.

Response:

We appreciate the reviewer's suggestion. We have now included the mirDeep2 PDF output for all read distributions of the novel miRNAs. Each output is quite lengthy and as such, have included them as revised Supplementary Fig. 9-13.

Reviewer #2 (Remarks to the Author):

The authors have presented a comprehensive profile of small non-coding RNAs (sncRNAs) in human preimplantation embryos. Within this, it was observed that miRNAs target known lineage-specific marker genes, showing a consistent pattern. Furthermore, correlations in expression were also demonstrated. The variations in the expression of modified tRNAs and isomiRs are novel and intriguing. Such a resource is unprecedented and represents highly significant information. It is suggested that for this study to be more suitable for publication in Nature Communications, it needs to go beyond just being a source of information and discussing correlations in expression changes. The study should also delve into deeper analysis.

Major Points:

1. The identification of C14MC as a hotspot in ICM-lineage expression is noted, but it would be beneficial to include information on the role it plays experimentally, similar to what has been done with C19MC in human ES cells.

Response: We agree with the reviewer and have attempted to identify the functional significance of C14MC as was done for C19MC¹³. Of note, the experiments performed with the C19MC paper in human ES cells resulted in an entire Nature Communications manuscript, indicative of the efforts (time and money) required to accomplish this. We were unfortunately unable to identify a bona fide promoter and succeed in generating a homozygous KO of this C14MC. We attempted to generate hESCs deficient for C14MC using a Cas9 RNP nucleofection protocol at the Pastor lab (McGill University), who have used this approach to successfully ablate nine different genes. Unfortunately, despite some success in generating

heterozygotes, we were never able to delete both copies of the allele (Please see Reviewer Figure 2 below). It may be that the region is essential in hESCs and thus we could not produce the deletant lines. Unfortunately, we are thus unable to provide functional data about the regulatory role of this locus at this time.

Reviewer Figure 2. A. Schematic of the C14MC locus. Sites of sgRNA and PCR primers are indicated **B.** PCR of bulk populations of human embryonic stem cells after targeting with sgRNA guide pairs. Note amplification from External primers after nucleofection. **C.** Examples of clonal lines with amplification from External primers, indicating deletion of at least one copy of the C14MC locus. **D.** Results of genotyping clonal hESC lines. **E.** Example clones from Round 3 (heterozygous cells nucleofected a second time). Note amplification of internal primer in all lines, indicating that no full knockout was generated.

2. Similarly, for the newly identified miRNAs by the authors, those that are adequately expressed in human ES cells should undergo functional validation. This would greatly enhance the quality of the paper.

Response:

We agree and thank the reviewer for this suggestion. We attempted to perform parallel experiments using our limited 8 cell embryos allocated for research. Unfortunately, due to our limited access to additional 8 cell human embryos, the poor survival rate associated with thawing non-vitrified cleavage stage embryos as a result of long term storage (stored prior to 2012) and use of slow freezing technique, there were too few replicates and too much variability in the results (survival and developmental progression in controls) to draw any conclusions. Instead, we utilised the mouse embryo to perform the validation experiments and demonstrated the function of novel_miRNA_4_chr4_29031 in preventing blastocyst formation (see Reviewer Figure 3A). Further, we performed scRNA-seq on cells collected from 8 cell stage novel miRNA -treated and control mouse embryos, and bioinformatic analysis identified both Sall1 and Med15 as potential downstream targets of this novel miRNA (see Reviewer Figure 3B,C). Indeed, 8 cell mouse embryos that were treated with 10 μ M novel_miRNA_4_chr4_29031 showed a significant impact on both Sall1 and Med15 protein expression, albeit in the opposite direction than we expected (see Reviewer Figure 3D,E). This speaks to the complexity underlying mRNA and protein regulation, particularly with miRNA involvement as others have highlighted in recent publications^{14,15}. We believe this further emphasises the importance of our work which has identified all miRNAs present and their dynamics during preimplantation development, as it establishes the foundational framework for which future studies can further explore the regulatory intricacies involving miRNAs. We have decided to not include these experiments in the manuscript however, as the expression of this novel miRNA could not be confirmed in mouse embryos. Similarly, we considered using hESC, however, functional validation in hESCs is not feasible given that expression of these novel miRNAs is embryo specific and not detected in either naive or primed human embryonic stem cell lines (expression checked by using small-seq data from GSE81287). This also precludes us from using stem cell based embryo models, such as blastoids would not serve as a good model. This analysis is important and further emphasises the importance of our dataset generated in this manuscript. Adequately benchmarking stem cell based models to their in vivo counterparts (as recommended by the ISSCR) is extremely important and given the lack of our understanding or expression of small ncRNAs in the human embryo, without our current dataset, this would be very difficult to do.

Reviewer Figure 3. **A.** Percentage of total mouse embryos treated with scrambled negative control or novel miRNA mimic (novel_miRNA_4_chr4_29031) that developed into blastocysts at E3.5. **B.** Volcano plot showing differentially expressed genes (DEGs) identified between novel miRNA-treated and control 8-cell mouse embryos. A false discovery rate (FDR) < 0.1 and $\log_2(\text{fold change}) > 0.25$ were used as thresholds. **C.** Violin plot showing Med15 and Sall1 expression in control and novel miRNA mimic 8-cell mouse embryos. **D.** Average nuclear fluorescence intensity (protein expression, measured in arbitrary fluorescence units (AFUs)) of Sall1 (Abcam, cat # ab31526) and Med15 (Santa Cruz, cat #sc-101185), in 8 cell mouse embryos treated with scrambled control versus 10 μ M novel miRNA mimic. Each data point represents the average intensity of all the cells in one embryo. **E.** Representative immunofluorescence images of control and mimic-treated 8 cell mouse embryos. DAPI = nuclear stain. BF = brightfield. ns = p > 0.05, **** = p < 0.0001.

3. The Discussion section currently feels redundant with a list of prior studies. It should be revised to include functional speculation and be more cohesive with the findings of the current study.

We thank you for this suggestion and have now reworked the discussion. We have left in reference to previous studies in some sections, as they do provide support for the speculated functional significance of specific miRNAs we have identified in the human embryo. Without directly performing many of these experiments in the human embryo, which at this time is not feasible for us given the lack of 8 cell (or earlier embryos), we do rely on speculation, which is drawn from those previous works.

Minor Points:

1. Line 58: Correct "scnRNA" to "sncRNA."

Response:

This has now been corrected, thank you. Please see line 59

2. Supplementary Figure 2B legend: Change "Morual" to "Morula."

Response:

This has now been corrected, thank you.

3. In Figures 3C and 3D, the IGV graphs for E5 (ICM-TE) have a shorter y-axis compared to others. This should be adjusted to match the length of the other axes.

Response:

We thank the reviewer for highlighting this. It has now been corrected (please see revised Fig. 3c and d). Since the "ICM-TE" track contains both positive and negative values, the heights of these two tracks have been doubled compared to the other tracks (the original height was set to the same value).

4. Line 196: Is this referring to Supplementary Figure 3?

Response:

Thank you for flagging this typo, which has now been corrected. Please see line 216, which now refers to revised Supplementary Fig. 4.

5. Supplementary Figure 6: The depiction of miRNA_33190 needs revision.

Response:

Thank you for highlighting this, we have now corrected it. Please see revised Supplementary Fig. 11.

Reviewer #3 (Remarks to the Author):

The paper Russell et al. provides a novel atlas of small non-coding RNAs in human preimplantation development.

The group generated a novel sncRNAs dataset along with an extensive bioinformatical analysis. They look into the abundance and kinetics of different type of sncRNAs across different stages of human embryonic day. They found sncRNAs corresponded to specific genomic hotspots, and enrichment of specific miRNAs clusters in different lineages (e.g. TE vs. ICM). They also identified six novel miRNAs, which seem to be conserved across species, and may have embryonic functions.

This dataset has the potential to be useful to the stem cell and developmental biologist community. However, I have two major concerns that I would like the authors to address. Firstly, the way the human embryos are staged is based on embryonic days. This has been shown to be flawed and it does not properly benchmark the embryos to the developmental time. In addition, I think it would add value to the paper to validate some of the identified sncRNAs with expression characterization and functional assays in human embryos (preferably) or other embryos, or in vitro models like human embryonic stem cells or human trophoblast stem cells.

Major points:

1. Embryonic stage evaluation of human embryos.

This paper stages the human embryos using embryonic days (E3 to E7). The authors use this method as this was used also in their previous paper (Petropoulos et al., Cell, 2016). However, since then, several other works pointed out the importance to stage the embryos based on morphokinetic analysis rather than embryonic days (Gerri et al., Nature, 2020; Regin et al., Human reproduction, 2023; Zhu et al., Elife, 2021; Firmin et al., Seminars in Cell and Dev Bio, 2021; and others). What is E3/E4/E5/E6/E7 equivalent to? How do the embryos look like at E3-E7? 8-cell? Morula? Cavitating morula? Expanded blastocyst? Using embryonic stages can work for species like the mouse, where pre-implantation embryos undergo a very stereotypic developmental progression, however frozen/thawed human embryos are very asynchronous. While there is no consensus in the field about the best way to stage the embryos, the authors should take inspiration from the papers mentioned above and make an effort to assess the human embryos morphologically before subjecting them to single cell dissociation. They should also make a scheme and show images of the representative embryos from E3 to E7, and describe in the methods how they do the staging of the embryos.

Response:

We apologise for the lack of clarity provided pertaining to the staging of our embryos and thank you for highlighting this. We agree that embryo staging is important and cannot solely rely on embryonic day; this also causes a lot of confusion in the field when published data is being leveraged for reanalysis. Human embryo development is dynamic and there is ambiguity, so in my expertise, though there is no consensus on the best way to benchmark the development, it is important to consider both the Meistermann staging and other parameters, which include developmental time, morphology and cell number. We had recognized this in the Petropoulos et al., 2016 paper and embryonic day was not solely the method used. We staged our embryos based on a combination of morphology/developmental progression and embryonic day, and as such had included

samples captured and categorised as early (16 cell precompaction) and late E4 (16 cells, compaction), early- mid- and late E5 (as opposed to only E5, in recognition of the importance of this developmental timepoint and also the variability within this window), E6 and E7, considering whether embryos were hatched and/or expanded, the size of blastocoel and quality of ICM/TE. We did not however explain our staging in detail at the time or included that information in the manuscript and broadly classified them by embryonic day. The collection of the samples included in the present manuscript started in 2017, prior to the Meistermann publication, so we initially staged the embryos using a combination of morphology, developmental progression and embryonic day. We did consider their criteria in this dataset, by aligning our notes with their staging criteria, however had not included this annotation in the initial manuscript as we do not examine changes in small ncRNAs with detailed granularity (for example, EPI, PE, mural, polar) since these could not be resolved and also lacked the power in our analysis when the embryos were stratified by B1, B2 etc staging. Further, we performed round robin comparisons, for the TE and ICM lineage between Meistermann staging and our staging, to determine differential expression of miRNAs (used as a proxy for all small ncRNAs), in the TE and ICM lineage and no major significant differences were observed. Finally, when examining the clustering of cells based on miRNA in an unbiased manner, we found that they aligned with embryonic day. Together, these analyses supported the use of staging the embryos based on embryonic day and morphology for this analysis. That said, we have now included a schematic (See revised Fig. 1a) to align the staging recommended by Meistermann and embryonic days and hope that this will provide clarity to future users who wish to leverage our dataset and again thank you for this important suggestion.

Further, included here we have highlighted the dataset stratified by our staging, which considered a combination of developmental time point, blastocoel size and cell number (please see revised Supplemental Fig. 2d-e and lines 155-157). We see that our staging generally aligns with the suggested Meistermann et al., staging. In our data, generally, 8C corresponded to the 8 cell embryo=E3, morula=E4, 16 cell embryo, B1=late E4 and early E5 with small blastocoel cavity emerging and all belonging to the prelineage category), B2=E5 (majority mid-E5), B3=late E5 and early E6, B4=mid E6 and early E7, blastocyst starting to expand, B5=expanded and hatching blastocyst, late E6 and mid E7, B6=hatched blastocyst at E7.

2. Further characterization and validation of key identified sncRNAs.

As mentioned throughout the manuscript, validating the role of sncRNAs is possible in both embryos and stem cell lines. I would like to ask the authors to do some experiments to support the possible involvement of some of the key sncRNAs identified. Can the authors look at the expression of some preferentially expressed miRNAs in TE and ICM via staining (e.g. RNA scope, RNA-ISH?) in human embryos? Could they assess their potential role using knockdown approaches in human embryos? If human embryos are not possible for such experiments, could they validate this in other species, e.g. mouse, if they are conserved? Otherwise, could some functional test be done in hESCs or hTSCs?

Response:

We agree and thank the reviewer for raising this point. Please see Responses to Reviewer 2, where we described the validation experiments and those that we have now incorporated

in the manuscript. Please also see Reviewer Figure 2 and 3, revised Fig. 7 and Supplementary Fig. 7 and 8.

Minor points:

3. Stirparo et al. (Development, 2018) reassessed the single cell transcriptomic data developed in human embryos. One of the datasets that they considered in their reanalysis was Petropolous et al., Cell, 2016. They raised the concern of possible TE over-representation in the Petropolous et al. dataset, and suggested that possibly there was incomplete immunolysis and ICM recovery in the embryos used in this study. I am wondering whether the authors took into account the re-analysis of Stirparo et al. when using the Petropolous et al. dataset in the current work. More importantly, do the authors performed ICM immunosurgery in this paper as well? I would like to ask them to clearly state this in the method and elaborate about the protocol used as much as possible, and not only cite their previous references.

Response:

Stiparo et al.¹⁶, had incorrectly interpreted the Petropoulos et al manuscript. We had stated that immunosurgery was performed to enrich the ICM population, not to obtain a pure ICM, as that was not the intended goal. When picking single cells from a human embryo, there is a bias toward collecting a disproportionate number of TE cells. So to circumvent this, we performed immunosurgery, to enrich but did not aim to obtain a perfect, pure ICM ‘pearl’. Interestingly, very recently, the Nicols lab has published a paper that demonstrated the activation of Gata3+ cells in ‘ICM’¹⁷. Nonetheless, in our current manuscript, we did not need to perform immunosurgery to effectively obtain an enriched ICM population as our clinical collaborators have access to laser dissection which allows the segregation of TE from the ICM. Further, and importantly, cell identity in this paper was confirmed by using the miRNA-mRNA co-sequencing approach where ICM and TE populations were defined by using well known gene signatures corresponding to each cellular lineage and then miRNAs associated with each lineage. Please see Methods lines 599-605.

4. I would like to ask if the authors could add the human zygote in their sample pool to have a complete view of the sncRNA dynamics.

Response:

This is a great suggestion and we would love to have been able to perform these experiments. Unfortunately in Canada we do not have access to human zygote embryos for research purposes. The earliest time point we have access to is E3 (8 cell stage). As clinical practice has shifted over the years, in light of higher rate of transfer success following vitrification at E5, access to even E3 embryos are extremely rare now. That said, there has been one fairly recent paper from a group in Finland that has included small ncRNA profiles in the zygote¹²; however their data is not available. Moreover, currently, there are no datasets that have looked at the profiles of all small ncRNAs in non-human primate embryos (or any other species). This is part of what makes our study a novel and important resource for the field, where future studies can focus on a comparative biology approach.

5. Is there some analysis that the authors could do to look into primate-specific scnRNAs?

Response:

This would be extremely interesting. As mentioned above, there are very limited studies that have examined all small ncRNAs in preimplantation embryos. To our knowledge, we are the first to perform such analysis throughout human preimplantation development and it does not exist in other species. Further, performing comparisons of datasets in mouse or other species from other platforms including microarray to this sequencing method would be confounded by technical noise and could lead to false conclusions. Finally, the majority of studies that have been performed in embryos have been done in bulk, thereby not necessarily capturing the ICM and TE specific profiles of all small ncRNAs, including miRNAs. That said, we were able to find one paper in non-human primate that focused solely on miRNAs⁵ (please see Reviewer Figure 4 and revised Supplementary Data 2 and 5). We have not included this analysis in our manuscript, as we are not entirely sure of the source and validity of their annotations. For example, C19MC is a well known primate specific cluster, yet when we examine the individual miRNAs associated with this cluster, this paper reports one as non-primate specific. Nonetheless, we have included a column in our revised Supplementary Data 2 and 5 to include species specificity. Please also see Methods, lines 657-661.

Reviewer Figure 4. **A.** Proportion of miRNAs associated with lineage markers grouped by species specificity. **B.** Proportion of miRNAs stratified by different clusters of ICM trajectory-related miRNAs. **C.** Proportion of miRNAs stratified by different clusters of TE trajectory-related miRNAs. **D.** Proportion of miRNAs stratified by developmental lineages and days. Colours indicate whether the miRNAs are primate-specific, non-primate-specific, or uncertain, respectively.

Response References

1. Liu, W. & Wang, X. Prediction of functional microRNA targets by integrative modeling of microRNA binding and target expression data. *Genome Biol.* **20**, 18 (2019).
2. Chen, Y. & Wang, X. miRDB: an online database for prediction of functional

- microRNA targets. *Nucleic Acids Res.* **48**, D127–D131 (2020).
3. Moradi, S. *et al.* Time-resolved Small-RNA Sequencing Identifies MicroRNAs Critical for Formation of Embryonic Stem Cells from the Inner Cell Mass of Mouse Embryos. *Stem Cell Rev Rep* **19**, 2361–2377 (2023).
 4. Wang, Y. *et al.* microRNAs Regulating Human and Mouse Naïve Pluripotency. *Int. J. Mol. Sci.* **20**, (2019).
 5. Hu, H. Y. *et al.* Evolution of the human-specific microRNA miR-941. *Nat. Commun.* **3**, 1145 (2012).
 6. Ziegenhain, C., Hendriks, G.-J., Hagemann-Jensen, M. & Sandberg, R. Molecular spikes: a gold standard for single-cell RNA counting. *Nat. Methods* **19**, 560–566 (2022).
 7. Fu, Y., Wu, P.-H., Beane, T., Zamore, P. D. & Weng, Z. Elimination of PCR duplicates in RNA-seq and small RNA-seq using unique molecular identifiers. *BMC Genomics* **19**, 531 (2018).
 8. Li, J., Zhang, Z., Zhuang, Y., Wang, F. & Cai, T. Small RNA transcriptome analysis using parallel single-cell small RNA sequencing. *Sci. Rep.* **13**, 7501 (2023).
 9. Hagemann-jensen, M., Abdullayev, I. & Sandberg, R. Small-seq for single-cell small RNA sequencing. *Nat. Protoc.* 1–61.
 10. Asami, M. *et al.* Human embryonic genome activation initiates at the one-cell stage. *Cell Stem Cell* **29**, 209–216.e4 (2022).
 11. Sha, Q.-Q. *et al.* Dynamics and clinical relevance of maternal mRNA clearance during the oocyte-to-embryo transition in humans. *Nat. Commun.* **11**, 4917 (2020).
 12. Paloviita, P. *et al.* Small RNA expression and miRNA modification dynamics in human oocytes and early embryos. *Genome Res.* **31**, 1474–1485 (2021).
 13. Kobayashi, N. *et al.* The microRNA cluster C19MC confers differentiation potential into trophoblast lineages upon human pluripotent stem cells. *Nat. Commun.* **13**, 3071 (2022).

14. Tarbier, M. *et al.* Landscape of microRNA and target expression variation and covariation in single mouse embryonic stem cells. *bioRxiv* 2024.03.24.586475 (2024) doi:10.1101/2024.03.24.586475.
15. Freimer, J. W., Hu, T. J. & Blelloch, R. Decoupling the impact of microRNAs on translational repression versus RNA degradation in embryonic stem cells. *Elife* **7**, (2018).
16. Stirparo, G. G. *et al.* Integrated analysis of single-cell embryo data yields a unified transcriptome signature for the human pre-implantation epiblast. *Development* **145**, (2018).
17. Corujo-Simon, E. *et al.* Human trophectoderm becomes multi-layered by internalization at the polar region. *Dev. Cell* (2024) doi:10.1016/j.devcel.2024.05.028.

REVIEWER COMMENTS

Reviewer #1 (Remarks to the Author):

With a single exception (see below) the authors have addressed all of my concerns. In particular, it is comforting that their analyses yield substantially more concordant than discordant miRNA-target pairs, and the functional characterization of miR-381-3p in mouse embryos is a valuable addition to the study.

When sequencing small RNAs from rare and valuable samples such as early human embryos, it makes sense to predict novel miRNAs. However, from the graphics of the six novel miRNA candidates, it appears that only two candidates, novel_miRNA_4_chr4_29031 and novel_miRNA_4_chr15_13546, have features that support their status as such. The remaining four sequences lack passenger strands, which are in these days with sensitive sequencing methods really necessary for putting forward a sequence as a miRNA candidate. Conservation of the seed sequence only counts as support if the seed is conserved in closely related species and in an evolutionary pattern that make sense (e.g. not present in humans and nematodes and in no other species). Therefore, the remaining four sequences should really be removed from the manuscript, or alternatively entirely delegated to the supplementary and not mentioned in the main manuscript.

Reviewer #2 (Remarks to the Author):

The authors have appropriately addressed the points raised during the review process and have validated the functionality of miRNA in mouse blastocysts. As a result, this paper suggests the importance of miRNA and other small RNAs in human embryogenesis. Its comprehensive data will serve as a valuable resource for many researchers, and I consider it suitable for publication.

Reviewer #3 (Remarks to the Author):

The authors have answered all the concerns I had.
I think the revised manuscript is suitable for publication in Nature Communications.

Dear Reviewers,

We thank all reviewers for the very quick turnaround and for deeming our revisions satisfactory. We have now addressed Reviewer #1's additional comment.

REVIEWER COMMENTS

Reviewer #1 (Remarks to the Author):

When sequencing small RNAs from rare and valuable samples such as early human embryos, it makes sense to predict novel miRNAs. However, from the graphics of the six novel miRNA candidates, it appears that only two candidates, novel_miRNA_4_chr4_29031 and novel_miRNA_4_chr15_13546, have features that support their status as such. The remaining four sequences lack passenger strands, which are in these days with sensitive sequencing methods really necessary for putting forward a sequence as a miRNA candidate. Conservation of the seed sequence only counts as support if the seed is conserved in closely related species and in an evolutionary pattern that make sense (e.g. not present in humans and nematodes and in no other species). Therefore, the remaining four sequences should really be removed from the manuscript, or alternatively entirely delegated to the supplementary and not mentioned in the main manuscript.

Response: We thank the reviewer for pointing this out. We have now removed mention of the four novel miRNAs from the main text. As all predicted novel miRNAs were already as Supplemental Figures, we have not modified the figures.